# CEBPA restricts alveolar type 2 cell plasticity during development and injury-repair

**Dalia Hassan[1,2] & Jichao Chen [1,3]** ✉

Cell plasticity theoretically extends to all possible cell types, but naturally decreases as cells differentiate, whereas injury-repair re-engages the developmental plasticity. Here we show that the lung alveolar type 2 (AT2)-specific transcription factor (TF), CEBPA, restricts AT2 cell plasticity in the mouse lung. AT2 cells undergo transcriptional and epigenetic maturation postnatally. Without CEBPA, both neonatal and mature AT2 cells reduce the AT2 program, but only the former reactivate the SOX9 progenitor program. Sendai virus infection bestows mature AT2 cells with neonatal plasticity where *Cebpa* mutant, but not wild type, AT2 cells express SOX9, as well as more readily proliferate and form KRT8/CLDN4+ transitional cells. CEBPA promotes the AT2 program by recruiting the lung lineage TF NKX2-1. The temporal change in CEBPA-dependent plasticity reflects AT2 cell developmental history. The ontogeny of AT2 cell plasticity and its transcriptional and epigenetic mechanisms have implications in lung regeneration and cancer.

Cell plasticity, semantically defined as a cell's ability to become a different cell type, determines the resilience and reaction of cells to genetic and environmental perturbations. Theoretically, nearly every cell has the same DNA blueprint to become any other cell type, as demonstrated by reprogramming of fibroblasts into pluripotent stem cells[1]. In tissues, more plasticity is desirable for progenitor/stem cells to fuel their more differentiated lineages and for direct cell fate switch during trans-differentiation, but can be hijacked during tumorigenesis[2]. Less plasticity is associated with developmental differentiation toward specialized physiology, as well as with ageing[3].

During development, progenitors gradually confine themselves first to particular germ layers, then anteroposterior dorsoventral locations and organs, and finally cell types within, gaining cell-type specificity and losing plasticity – apparently two sides of the same coin. Analogous to evolutionary speciation, this developmental history sets the time of divergence among cell types and the rates afterwards—trajectories potentially predictive of the direction and extent of cell plasticity. The graduality of development is punctuated by points of no return, beyond which cells cannot revert to their developmental ancestors, as illustrated by closure of the neonatal regenerative window for the mammalian heart[4]. These points of no return often differ

from the points of cell fate specification, highlighting the importance of post-specification maturation and the discordance between the said two sides of the specificity-plasticity coin. Therefore, factors traditionally studied as cell-fate promoting need to be separately evaluated for plasticity restricting. Such studies of reactivating developmental plasticity and overcoming points of no return can inform tissue regeneration and are medically important.

The definitional dependence of cell plasticity on cell types results in operational difficulties, as the spectrum of variable states within a cell type often bleeds into related cell types, especially during development and upon injury, introducing uncertainty in cell types and thus plasticity. However, the continuum of cell types can be objectively quantified by molecular profiling of the transcriptome, epigenome, proteome, etc., especially when done on a single-cell level. Accordingly, this study defines cell plasticity as gains in molecular features characteristic of other cell types.

Building on our prior transcriptional and epigenetic studies of the mouse lung epithelium, this study focuses on the alveolar type 2 (AT2) cells that originate from SOX9 embryonic progenitors and become facultative stem cells, which secrete pulmonary surfactants at baseline, but self-renew and give rise to the gas-exchanging alveolar type 1 (AT1)

[1]Department of Pulmonary Medicine, the University of Texas MD Anderson Cancer Center, Houston, TX 77030, USA. [2]The University of Texas MD Anderson Cancer Center UTHealth Graduate School of Biomedical Sciences, Houston, TX 77030, USA. [3]Department of Pediatrics, Perinatal Institute Division of Pulmonary Biology, University of Cincinnati and Cincinnati Children's Hospital Medical Center, Cincinnati, OH 45229, USA. ✉e-mail: jichao.chen@cchmc.org

cells upon injury[5,6]. This life cycle of AT2 cells provides an experimental paradigm to probe the ontogeny of cell plasticity and associated transcriptional regulators. We show that AT2 cells use a cell-type-specific transcription factor (TF) CEBPA to restrict their plasticity at neonatal and mature stages, but reactivate the developmental plasticity upon respiratory virus infection. Mechanistically, CEBPA recruits the lung lineage TF NKX2-1 to promote the AT2 cell program and indirectly represses the SOX9 progenitor program, within the confinement of their developmental history.

## Results

### Postnatal transcriptomic and epigenomic maturation of AT2 cells is separate from their embryonic specification

To explore how cell plasticity can be shaped by development, we first delineated the associated molecular progression of AT2 cells using scRNA-seq and scATAC-seq. Mouse lung epithelial cells profiled by scRNA-seq at 12-time points spanning from embryonic day (E) 14.5 to 15-week adult stages, clustered by time and cell types (Fig. 1A, Supplementary Fig. 1). AT2 cells clustered separately from AT1 cells only by E18.5, indicating measurable specification from their E14.5 and E16.5 SOX9 progenitors (Fig. 1A). Nascent AT2 cells at E18.5 were distinct from their neonatal and adult counterparts, suggesting maturation following specification, which was supported by Monocle trajectory analysis (Fig. 1B). Eighty-two genes specific to 15-week AT2 cells, in comparison to 15-week AT1 cells, were designated as early versus late AT2 genes based on their presence versus absence of expression in >25% of E16.5 SOX9 progenitors[7] (Fig. 1C). Ten out of the 41 late AT2 genes, including immune response genes (*Lyz2*, *Lrg1*, *Chil1*, and *H2* paralogs), only reached maximal expression (>4-fold increase from E18.5 to 15-week) during postnatal maturation (Fig. 1C).

Supporting this transcriptional post-specification maturation, scATAC-seq sampling of 5 developmental time points from E16.5 to 9-week adult revealed epigenomic maturation of AT2 cells that occurred after their E18.5 specification (PCA plot in Fig. 1D). Accordingly, we classified differential ATAC peaks into lost and gained groups, each with early and late subgroups separated at the E18.5 time point of specification (Fig. 1D). Although ATAC peaks often correlate with RNA transcripts and may act over long-range and on high-order chromatin structure, they measure individual regulatory regions without averaging over the whole gene, predict regulatory TF motifs, and bypass the confounding issue of perdurance in RNA-seq. As shown in Fig. 1D, the early lost peaks coincided with AT2 specification at E18.5, were near progenitor and AT1 genes (e.g., *Adamts18* and *Clic5*, respectively; see, Supplementary Fig. 2 for representative genomic snapshots; same below), and contained SOX and TEAD motifs, likely reflecting the termination of the SOX9-mediated progenitor program and the low level of YAP/TAZ/TEAD-mediated AT1 program in E16.5 progenitors[7,8]. The late lost peaks decreased postnatally, were near stem cell genes (e.g., *Klf4*, *Id4*, *Etv6*, and *Hif3a*)[1,9–11], and contained FOXA and NKX motifs, correlating with the postnatal decrease in AT2 cell proliferation and potential progenitor-specific functions of FOXA2 and NKX2-1. Conversely, the majority of differential peaks (19,258 out of 30,410; 63%) were in the early gained subgroup and were near AT2 genes (e.g., *Lyz1*) and enriched for CEBP and NKX motifs, which were examined in detail in this study. Last, the late gained peaks followed the RNA kinetics of AT2 cell maturation (Fig. 1C), were near the corresponding genes (e.g., *H2-Aa* and *Cd74*), and contained the AP-1 motif. Supporting the epigenomic distinction between specification and maturation, each of the 4 groups was associated with distinct biological pathways and unsupervised clustering of top 100 variable motifs predicted from scATAC-seq revealed lost, early gained, and late gained groups, dominated by SOX and TEAD, CEBP, and AP-1 motifs, respectively (Supplementary Fig. 3A, B).

We reasoned that, while the gain in chromatin accessibility reflected AT2 cell differentiation, the concurrent loss in accessibility

not only indicated the conclusion of prior cell fates, but also predicted available fates when AT2 cells became more plastic. Accordingly, TFs promoting AT2 cell differentiation might also restrict AT2 cell plasticity. Of particular interest was CEBPA, whose motif was enriched among the early gained peaks, as well as our published AT2-specific NKX2-1 ChIP-seq peaks[8] (Fig. 1D). Compared to other CEBP family members, *Cebpa* was specific to AT2 cells and reached maximal expression coinciding with AT2 specification (Supplementary Fig. 3C, D). Supporting this, CEBPA was absent on the protein level at E14.5 when branch tips consisted of SOX9+ progenitors and had weak diffuse expression in occasional cells at E16.5, likely corresponding to spatially asynchronous onset of alveolar differentiation and consistent with prior reports[12,13]; from E18.5 on, CEBPA was expressed in a subset of epithelial cells that were cuboidal, SOX9- HOPX- LAMP3 + , and thus nascent AT2 cells (Fig. 2A, Supplementary Fig. 3E, F). Taken together, our time-course transcriptomic and epigenomic roadmap of AT2 cell development expand on published datasets[8,14,15], highlighting the sequential specification and maturation of AT2 cells and implicating CEBPA in their differentiation and plasticity.

### CEBPA promotes AT2 and suppresses progenitor programs in neonatal AT2 cells

Although CEBPA had been shown to promote both AT1 and AT2 cell differentiation in embryonic lungs[16], its role in subsequent AT2 cell maturation and plasticity was unclear. Accordingly, we generated an inducible AT2-specific knockout model *Cebpa*^F/F^; *Sftpc*^CreER/+^; *Rosa*^Sun1GFP/+^. To target AT2 cells shortly after specification, we induced Cre-recombination at a neonatal stage (P2) with an efficiency of 99% and specificity of 96% (1310 GFP+ cells from 3 mice) and deleted CEBPA in AT2 cells with an efficiency of 88% (1951 GFP+ cells from 3 mice), without affecting its normal expression in alveolar macrophages (Fig. 2B). By P9, *Cebpa* mutant AT2 cells had a drastic decrease in LAMP3 and a loss of IL33, both AT2 markers, compared to adjacent escapers of Cre-recombination or AT2 cells in the littermate control (Fig. 2B, Supplementary Fig. 4A). Transmission electron microscopy showed that cuboidal/columnar epithelial cells in the *Cebpa* mutant lung often lacked lamellar bodies, a defining feature of AT2 cells, but still had characteristic apical microvilli (Fig. 2C, Supplementary Fig. 4B).

Our previous study of the CEBPA equivalent in AT1 cells, YAP/TAZ/TEAD, showed activation of the alternative alveolar program[8] and thus prompted us to examine the AT1 program in *Cebpa* mutant AT2 cells. To our surprise, only a small fraction of mutant AT2 cells expressed an AT1 marker HOPX (12.5% of 2049 GFP+ cells from 3 mice), lost LAMP3, and were no longer cuboidal as outlined by E-Cadherin, compared to a baseline of 1.2% in the control (1353 GFP+ cells from 3 mice) that possibly resulted from driver non-specificity/leakiness and normal neonatal conversion of AT2 to AT1 cells (Fig. 2D). Instead, we noticed adjoining *Cebpa* mutant cells reminiscent of SOX9 progenitors at embryonic branch tips (Supplementary Fig. 4C). Remarkably, 80% of GFP+ cells in the mutant lung (1202 GFP+ cells from 3 mice) expressed SOX9, compared to 0.7% in the control (1370 GFP+ cells from 3 mice) (Fig. 2E). Like SOX9 progenitors, *Cebpa* mutant cells were also much more proliferative (KI67+ in 25% 3414 GFP+ cells from 3 mice, compared to 6% 2737 GFP+ cells from 3 mice in the control), likely contributing to their clustering (Fig. 2E, F). Therefore, without CEBPA, neonatal AT2 cells reduce their AT2 program and gain plasticity toward SOX9 progenitors and, to a much lesser extent, AT1 cells.

### Single-cell multiome defines CEBPA-dependent neonatal AT2 cell program and plasticity

To fully characterize the CEBPA-dependent changes, we FACS-purified E-Cadherin+ epithelial cells from our neonatal *Cebpa* mutant and littermate control lungs and performed single-cell multiome for concurrent profiling of their transcriptomes and epigenomes

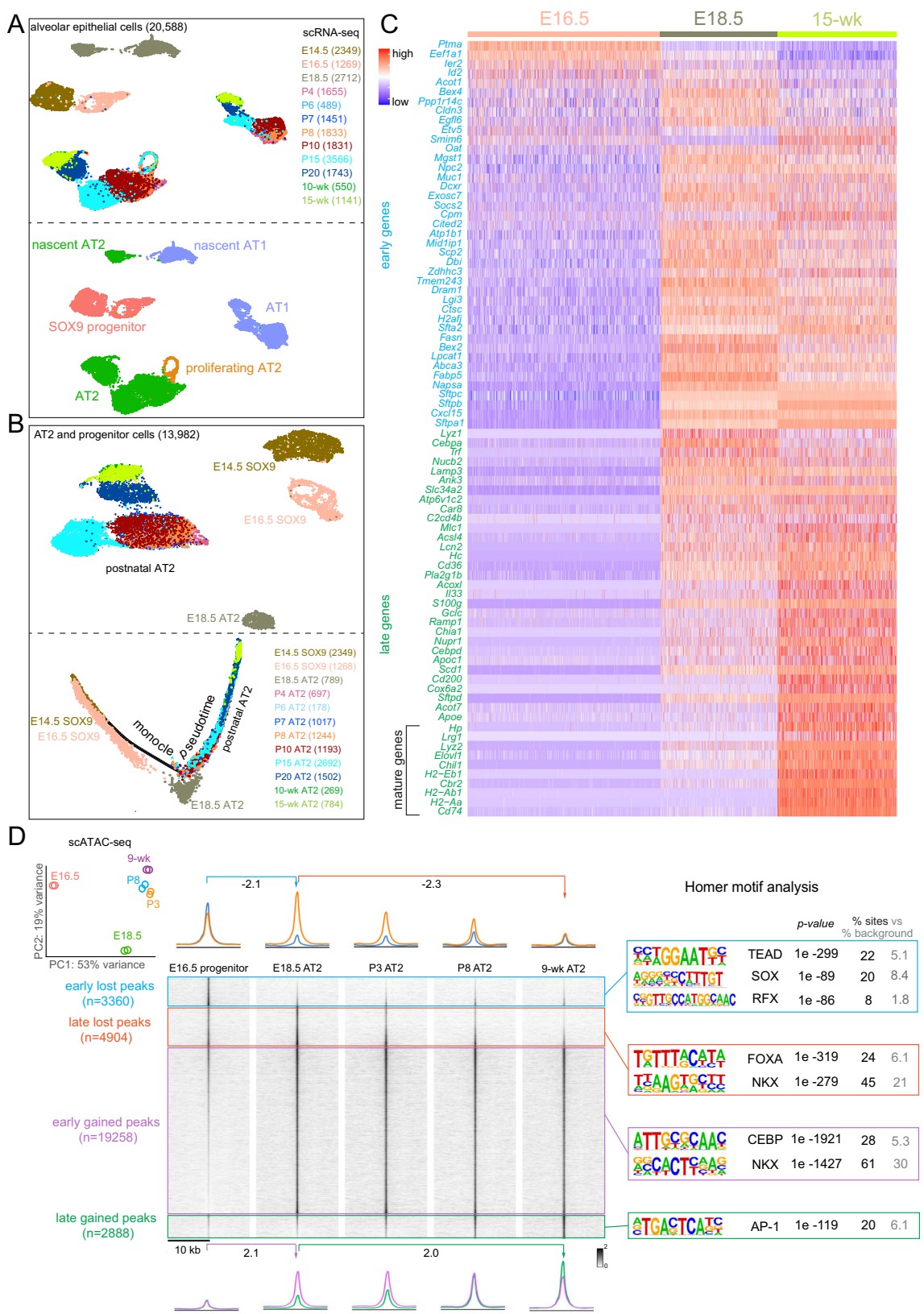

(Supplementary Fig. 5A). On the combined single-cell UMAPs, club, ciliated, and AT1 cells from control and mutant lungs formed super-imposed clusters, suggesting minimal changes, as they were not targeted by *Sftpc*[CreER] and lacked the GFP transcript from *Rosa*[Sun1GFP] (Fig. 3A, B, Supplementary Fig. 5B). In contrast, while 3.7% (267 out of 7047 cells) AT2 cells from the mutant lung were intermixed with those

from the control and still expressed *Cebpa*, consistent with them being escapers of deletion, the rest formed a separate *Cebpa*⁻ cluster (mutant AT2), as well as a proliferative cluster mainly made of cells from the mutant lung as predicted by KI67 immunostaining (Fig. 3A, B). A marker *Lamp3* and a 100-gene signature score for AT2 cells were reduced in mutant AT2 cells, whereas a marker *Sox9* and a 119-gene

**Fig. 1 | AT2 cells undergo transcriptomic and epigenomic maturation postnatally, separate from their embryonic specification. A** Aggregated scRNA-seq UMAPs of alveolar epithelial cells of wild type lungs from 12-time points color coded by time (top) or cell type (bottom). Cell numbers are in parenthesis. Each time point consists of at least 2 mice profiled as one sample. **B** Top: UMAP of AT2 and SOX9 progenitor cells subset from in (**A**), color-coded by time, and their monocle pseudotime analysis (bottom) showing molecular progression from E14.5 SOX9 progenitors through E18.5 nascent AT2 to 15-wk mature AT2 cells. Cell numbers are in parenthesis. **C** Expression heatmap of 82 AT2-specific genes,

classified as early if present in at least 25% of cells at E16.5. The remaining late genes are considered mature if the fold increase from E18.5 to 15-wk is more than 4. **D** Principal component analysis (PCA) of scATAC-seq pseudobulk duplicates showing the distinct epigenome of E18.5 nascent AT2 cells. ScATAC-seq heatmaps and profile plots were categorized and color-coded as early lost (E16.5 vs E18.5), late lost (E18.5 vs 9-wk), early gained (E16.5 vs E18.5) and late gained (E18.5 vs 9-wk) and their Homer motif analysis. Peak numbers are in parenthesis. Each time point consists of at least 2 mice profiled as one sample.

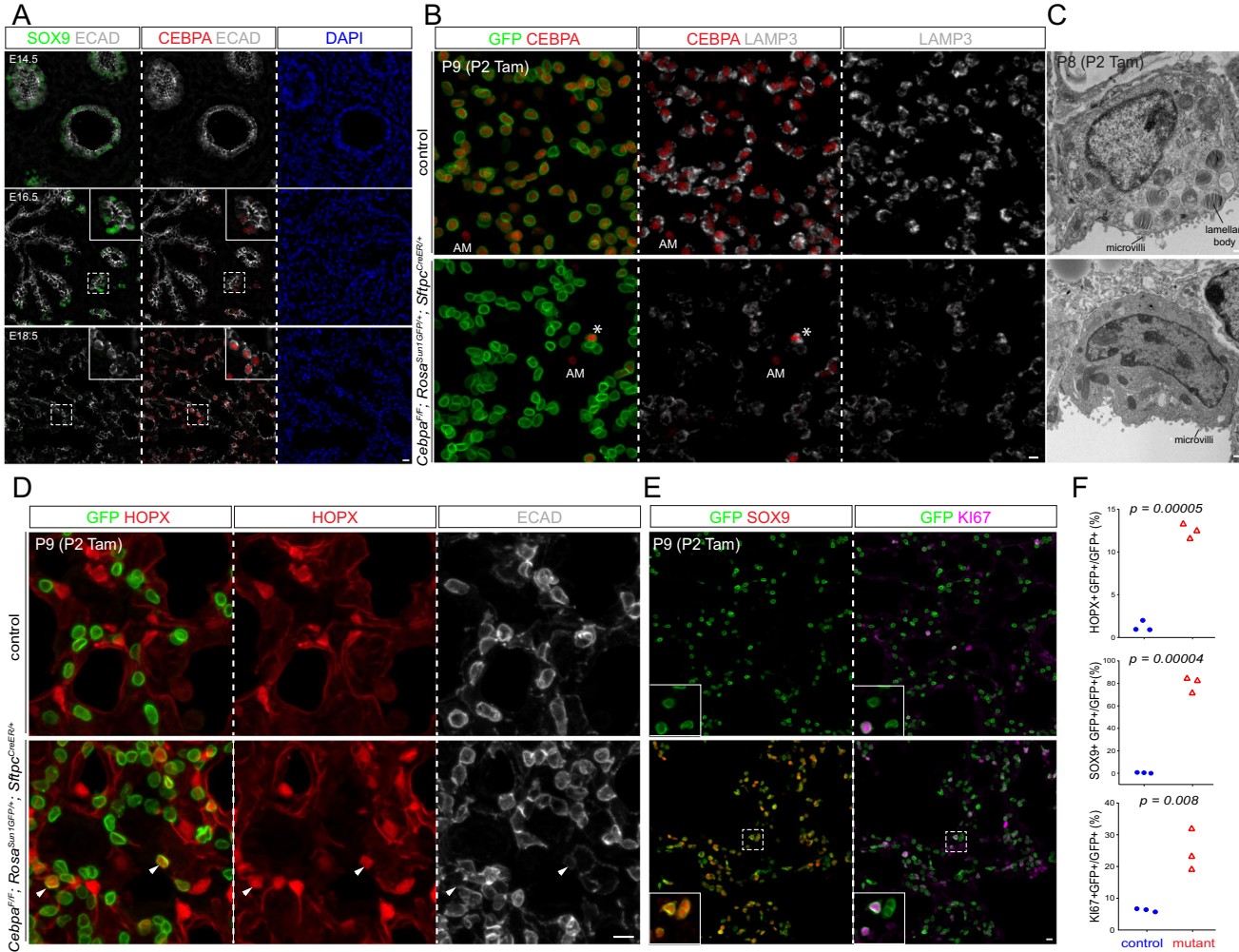

**Fig. 2 | CEBPA promotes AT2 and suppresses progenitor programs in neonatal AT2 cells. A** Confocal images of immunostained wild type lungs show little CEBPA expression at E14.5 and E16.5 when branch tips are dominated by SOX9 progenitors but CEBPA expression in cuboidal cells outlined with E-Cadherin (ECAD) at E18.5 (*n* = 3 mice each). **B** Confocal images of immunostained neonatal AT2-specific *Cebpa* mutant and littermate control lungs showing loss of CEBPA in GFP+ recombined cells (asterisk: escaper), without affecting its expression in alveolar macrophages (AM) in the airspace, and reduced LAMP3. Tam, 250 µg tamoxifen. Images are representative of at least three lungs (same for subsequent immunostainings). **C** Transmission electron microscopy (TEM) images show a reduction in

lamellar bodies in mutant AT2 cells without affecting their apical microvilli (*n* = 2 mice each). Tam, 250 µg tamoxifen. See Supplementary Fig. 4B for more examples and quantification. **D** Confocal images showing lineage labeled mutant AT2 cells expressing an AT1 marker HOPX and no longer cuboidal (ECAD outline) (arrowhead). **E** Confocal images showing lineage labeled mutant AT2 cells ectopically expressing a progenitor marker SOX9 and a proliferation marker KI67. **F** Quantification of (**D**) and (**E**). Each symbol represents one mouse from littermate pairs. *P* values were calculated using two-tailed Student's *t* test. Scale: 10 µm for all except for (**C**) 1 µm.

signature score for SOX9 progenitors were increased (Fig. 3C). The mutant AT2 cluster extended toward the AT1 cell cluster, forming a bridge that was GFP+ and thus descendant of recombined AT2 cells (Fig. 3C). This bridging population was specific to the mutant lung and expressed a marker *Hopx* and a 100-gene signature for AT1 cells, but still clustered separately from normal AT1 cells, possibly due to their AT2 cell origin and limited time for AT1 differentiation after *Cebpa*

deletion. RNA velocity analysis confirmed this predicted trajectory bridging AT2 and AT1 cells specifically in the mutant (Supplementary Fig. 5B). Considering the observed HOPX immunostaining in LAMP3-non-cuboidal cells (Fig. 2D, Supplementary Fig. 5C), we named this bridging population HOPX + AT1-like cells. Differential expression analysis of *Cebpa* mutant versus control AT2 cells confirmed downregulation of AT2 genes (e.g., *Lyz2*, *Lyz1*, *Sftpb*, and *Il33*) and

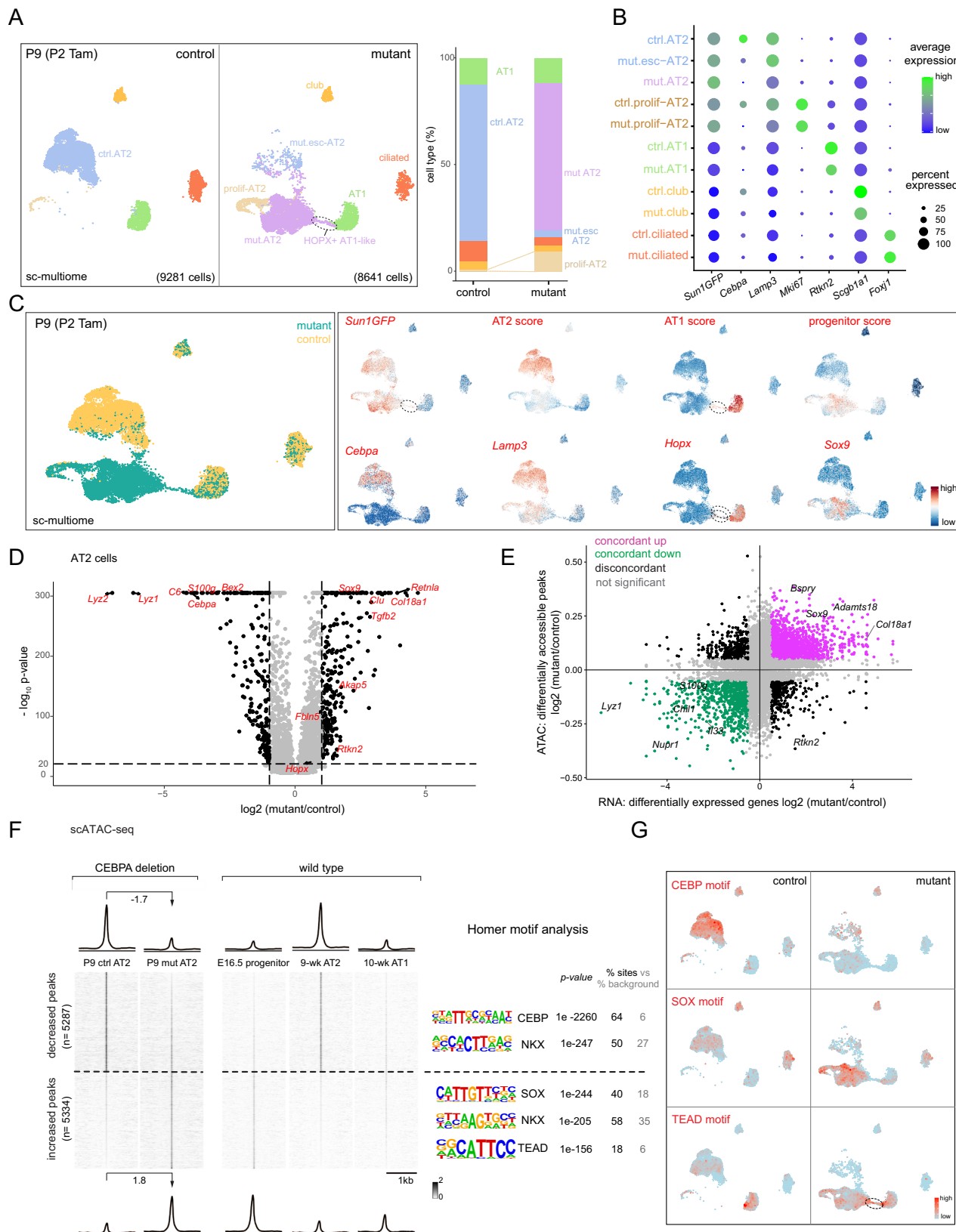

upregulation of progenitor (e.g., *Sox9*, *Clu*, and *Col18a1*) and AT1 genes (e.g., *Akap5*, *Fbln5*, and *Rtkn2*), although some surfactant genes including *Sftpc* were less reduced – unlike their near-absence upon embryonic *Cebpa* deletion[16], possibly due to RNA perdurance or redundant transcriptional activation (Fig. 3D, Supplementary Fig. 5D, E). Relatedly, opposite to the phenotypes reported here and

given the AT2-restricted expression of CEBPA (Fig. 2A, Supplementary Fig. 3E, F), the defective AT1 cell differentiation in the pan-epithelial embryonic *Cebpa* mutant[16] was likely non-cell autonomous or due to potential toxicity associated with the Cre driver[17].

To explore the epigenetic mechanism of the transcriptional changes, we performed differential accessibility analysis of *Cebpa*

**Fig. 3 | Single-cell multiome defines CEBPA-dependent neonatal AT2 cell program and plasticity. A** Sc-multiome UMAPs of purified epithelial cells from *Cebpa* mutant and littermate control lungs color-coded by cell type (left) and the corresponding percentages (right). Esc, escaper; prolif, proliferating; Tam, 250 μg tamoxifen. See Supplementary Fig. 5A for the sorting strategy. Each sample includes 1 male and 1 female mouse profiled as one sample (same for subsequent sc-multiome experiments). **B** Dot plot showing the lineage marker (Sun1GFP), *Cebpa*, and cell type markers. See also feature plots in (**C**) and Supplementary Fig. 5B. **C** Sc-multiome UMAP color-coded for genotype (left) and feature plots of metagene scores (top) and representative genes (bottom). The circled population is specific to the mutant, GFP + , and expresses AT1 genes, thus labeled as HOPX + AT1-like

cells in (**A**). See Source Data for metagene lists. **D** Volcano plot (two-tailed, nonparametric Wilcoxon rank sum test) showing downregulation of AT2 genes and upregulation of progenitor genes in mutant AT2 cells (right) compared to control AT2 cells (left) defined in (**A**). **E** Scatter plot correlating changes in the accessibility of scATAC-seq peaks (y-axis) and scRNA-seq expression of their nearest genes (x-axis), color-coded as concordant or discordant as well as the directionality of change. **F** ScATAC-seq heatmaps and profile plots of decreased and increased peak sets in the mutant and associated log2 fold changes, as well as the corresponding scATAC-seq data in wild type cells and associated Homer motifs. **G** Feature plots of motif activity scores showing that the mutant has lower CEBP, and higher SOX and TEAD (circle) activities.

mutant versus control AT2 cells as pseudobulks and identified 10,621 differential peaks. Assigning each peak to its nearest gene, we found a high concordance (71%) between peaks and gene expression, with increases in both (39%) for progenitor genes including *Sox9, Bspry,* and *Adamts18* and decreases in both (32%) for AT2 genes including *Lyz1, Il33,* and *S100g* (Fig. 3E). Moreover, the 5,287 decreased peaks were normally more accessible in AT2 cells, in comparison to AT1 and SOX9 progenitor cells, and were enriched for CEBP and NKX motifs, consistent with *Cebpa* deletion (Fig. 3F, G). The 5334 increased peaks were normally more accessible in SOX9 progenitor or, to a lesser extent as expected from the small number of HOPX + AT1-like cells, AT1 cells, in comparison to AT2 cells, and were enriched for SOX and TEAD motifs, consistent with activation of progenitor and AT1 programs (Fig. 3F, G). Therefore, besides the more predictable role of CEBPA in promoting the AT2 program, marker and whole-genome analyses unexpectedly show that neonatal AT2 cells have the plasticity to revert to SOX9 progenitors when unconstrained by CEBPA. The 5334 increased peaks represent CEBPA-dependent plasticity of neonatal AT2 cells that will be explored later.

## CEBPA recruits NKX2-1 to promote the AT2 program and indirectly restricts the progenitor program

The identification of differential accessibility peaks that were largely concordant with differential gene expression prompted mechanistic analysis to link ATAC peaks to CEBPA chromatin binding. Given the enrichment in CEBP and NKX motifs (Fig. 3F) and the normal expression of NKX2-1 in the *Cebpa* mutant (Supplementary Fig. 5F), we used our published cell-type-specific ChIP-seq protocol[8] to perform CEBPA ChIP-seq on control AT2 cells at P2, the time of Cre-recombination, as well as NKX2-1 ChIP-seq on AT2 cells from control and *Cebpa* mutant lungs at P8 (Supplementary Fig. 6A, Fig. 4A). The 5287 decreased peaks, as exemplified by a peak near an AT2 gene *Il33*, were bound by CEBPA and NKX2-1; NKX2-1 binding decreased upon *Cebpa* deletion, suggesting recruitment of NKX2-1 by CEBPA. NKX2-1 binding at these sites was specific to AT2 cells but not E14.5 SOX9 progenitors despite NKX2-1 expression in both, suggesting that CEBPA was required to acquire AT2-specific NKX2-1 binding (Fig. 4A, B). The predicted CEBP and NKX motifs, but not SOX motif, were concentrated at the center of the decreased peaks (Fig. 4A). Furthermore, the average distance between CEBPA and NKX2-1 binding sites were 54 bp, consistent with proximity or even direct binding between CEBPA and NKX2-1, although such protein-protein interaction needed technically challenging biochemical studies of purified AT2 cells. The parallel changes in both chromatin accessibility and NKX2-1 binding during their developmental gain from progenitors to AT2 cells and loss upon *Cebpa* deletion (Figs. 3F, 4A diagram) does not establish the sequence of events nor rule out the possibility that despite the presence of adjacent CEBPA and NKX2-1 motifs, decreased NKX2-1 binding to these sites was secondary to chromatin closure due to some impact of CEBPA deletion elsewhere in the genome, calling for future systematic deletion of CEBPA binding sites and/or interference of CEBPA binding to them. Relatedly, the recruitment model (Fig. 4A diagram) highlights the new

chromatin binding specificity of NKX2-1 conferred by CEBPA, but does not require sequential binding of CEBPA and then NKX2-1 to the chromatin.

In contrast, the 5334 increased peaks, as exemplified by a peak near a progenitor gene *Acaca*, had little CEBPA binding and a slight increase in NKX2-1 binding upon *Cebpa* deletion, possibly attributable to its relocation to progenitor and AT1-specific sites without sequestration by CEBPA (Fig. 4A, B). Interestingly, NKX2-1 bound more at these sites in E14.5 SOX9 progenitors as well as AT1 cells compared to AT2 cells, suggesting that these progenitor-specific NKX2-1 binding sites normally lost NKX2-1 binding and were closed during alveolar differentiation, but were reopened upon *Cebpa* deletion (Fig. 4A). Alternatively, given the robust SOX9 expression and the prevalent SOX motif for these sites (Figs. 2, 3), CEBPA directly or indirectly repressed *Sox9*, which in turn initiated the progenitor program. Indeed, we identified a putative regulatory region 3′ to *Sox9* that was open in the SOX9 progenitors and reopened in the *Cebpa* mutant, a profile mirrored by NKX2-1 binding (Supplementary Fig. 7).

The link between CEBPA/NKX2-1 chromatin binding and differential accessibility peaks was also examined in the reverse direction. CEBPA and NKX2-1 binding sites in AT2 cells were categorized as co-bound and single-bound for each TF (Fig. 4C). Compared to NKX2-1 single-bound sites, CEBPA/NKX2-1 co-bound sites, had a greater decrease in NKX2-1 binding and accessibility upon *Cebpa* deletion, reinforcing the said recruitment model and implicating other regulators of NKX2-1 binding and accessibility at the NKX2-1 single-bound sites (Fig. 4D). The CEBPA single-bound sites had limited accessibility and NKX2-1 binding as well as limited changes, suggesting a minor impact of CEBPA on its own (Fig. 4D). Taken together, in AT2 cells, CEBPA recruits NKX2-1 to promote the AT2 program, but does not bind to and thus indirectly represses sites that remain plastic in neonatal AT2 cells.

## CEBPA maintains the AT2 program without affecting the progenitor program in mature AT2 cells

As the transcriptomic and epigenomic landscape of AT2 cells matured postnatally (Fig. 1), we posited that they would reinforce their gene regulatory network and exhibit less cell plasticity. To test this, we induced Cre-recombination in mature AT2 cells in >5-week old lungs and achieved 92% efficiency in deleting *Cebpa* (1919 GFP+ cells from 3 mice), again without affecting CEBPA expression in alveolar macrophages (Fig. 5A). As in the neonatal deletion model, mature *Cebpa* mutant AT2 cells downregulated LAMP3, lost IL33, and had fewer lamellar bodies (Fig. 5A, B, Supplementary Fig. 6B, C). However, they did not express SOX9 or KI67, suggesting a loss of cell plasticity toward SOX9 progenitors (Fig. 5A).

Single-cell multiome profiling of E-Cadherin+ epithelial cells from mature *Cebpa* mutant and littermate control lungs showed a transcriptional shift only in targeted GFP + AT2 cells (Fig. 5C, D). Notwithstanding additional heterogeneity including a *Lyz1*+ population in the control lung and a *Meg3*+ population in the mutant, possibly related to lung cancer and fibrosis[18,19], the most prominent change was

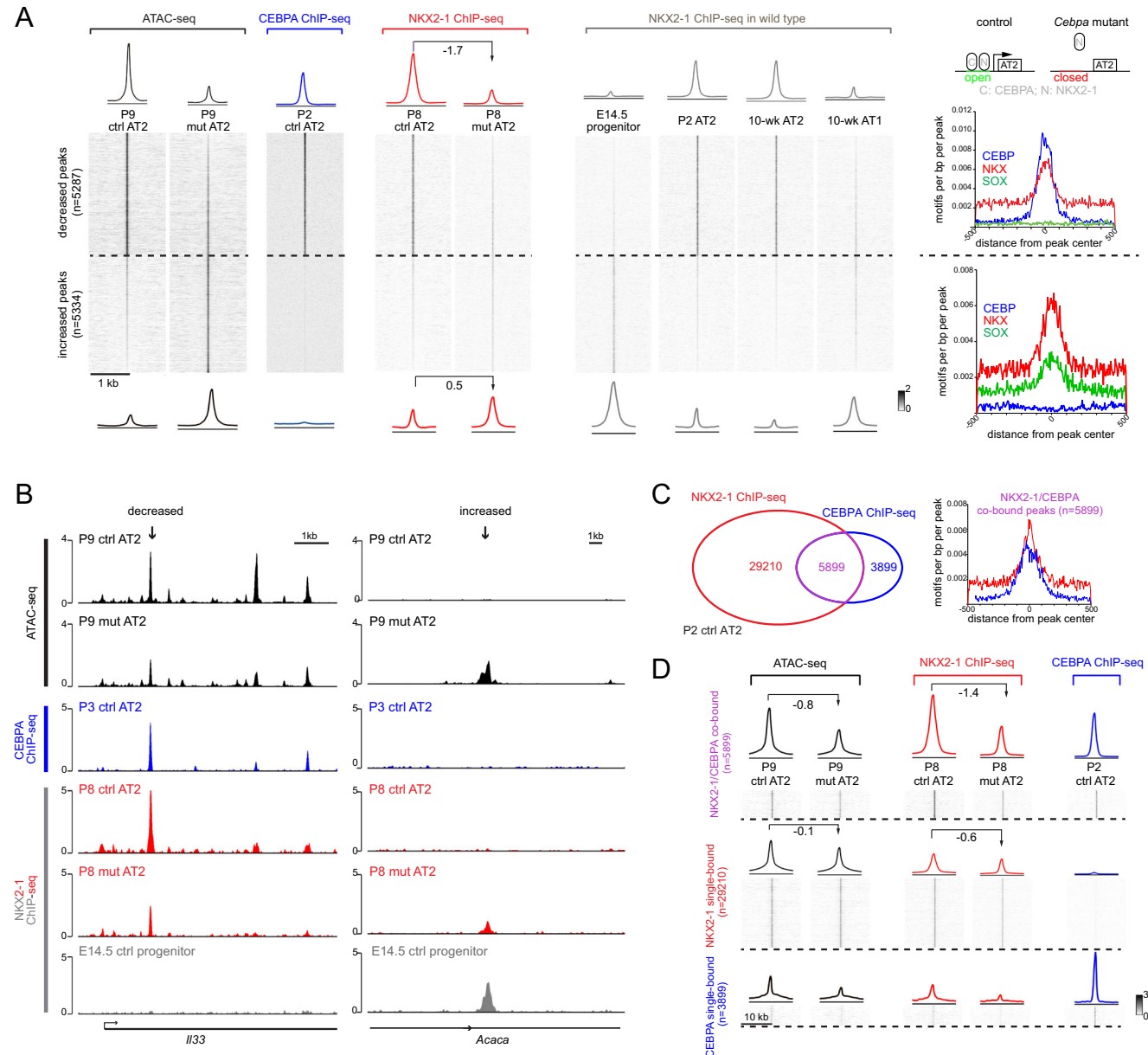

**Fig. 4 | CEBPA recruits NKX2-1 to promote the AT2 program and indirectly restricts the progenitor program. A** Heatmaps and profile plots of CEBPA and NKX2-1 binding for decreased and increased peak sets from Fig. 3F, as well as associated frequency distributions of CEBP, NKX, SOX motifs. CEBPA binds to decreased peaks but not increased peaks. NKX2-1 binding decreases (log2 fold change) for decreased peaks and increases (log2 fold change) for increased peaks in the mutant, corresponding to AT2 and progenitor/AT1-specific binding in wild type lungs. Diagram: a recruitment model, in which CEBPA normally recruits NKX2-1 to activate AT2 genes, whereas without CEBPA, NKX2-1 is released from AT2 genes and possibly relocates to progenitor and AT1 genes. Loss of NKX2-1 binding due to *Cebpa* deletion is associated with chromatin closure (open vs closed). See Supplementary Fig. 6A for nuclei sorting strategy. **B** Representative coverage plots of (**A**) showing a decreased peak near an AT2 gene *Il33*, and an increased peak near a progenitor gene *Acaca*. **C** Venn diagram showing NKX2-1 and CEBPA co-bound and single-bound peak sets in purified P2 AT2 cells (left) and frequency distributions of NKX and CEBP motifs for the co-bound peak set (right). **D** Heatmaps and profile plots for the 3 peak sets in (**C**) and associated log2 fold changes showing the largest decreases for the co-bound peak set.

downregulation of AT2 genes in *Cebpa*⁻ mutant AT2 cells, without activating progenitor genes or forming a proliferative population as in the neonatal lungs (Fig. 5C). Compared to the control lung, the *Cebpa* mutant lung had a larger population clustered near AT2 cells and expressing some but not all AT1 gene transcripts (Fig. 5C, D), although few HOPX+ cells were detected by immunostaining (Fig. 5A). Accordingly, we considered this population HOPXˡᵒʷ AT1-like cells to indicate their limited AT1 differentiation (Fig. 5C). The reduction in the AT2 program, no increase in the progenitor program and limited increase in the AT1 program were supported by differential gene expression analysis (Fig. 5E).

Compared to the neonatal model, differential accessibility analysis of *Cebpa* mutant versus control mature AT2 cells identified 2619 decreased peaks but only 692 increased peaks, suggesting less CEBPA-dependent cell plasticity than neonatal AT2 cells. These differential peaks were still 76% concordant with gene expression (Fig. 5F). The decreased peaks were AT2-specific, enriched for CEBP and NKX motifs, had CEBPA and NKX2-1 binding in control AT2 cells but decreased NKX2-1 binding in *Cebpa* mutant AT2 cells, and had NKX2-1 binding in normal AT2 but not progenitor nor AT1 cells, supporting the same recruitment model of NKX2-1 by CEBPA in mature AT2 cells (Fig. 5G, Supplementary Figs. 6D, 4A diagram). The few increased peaks had no

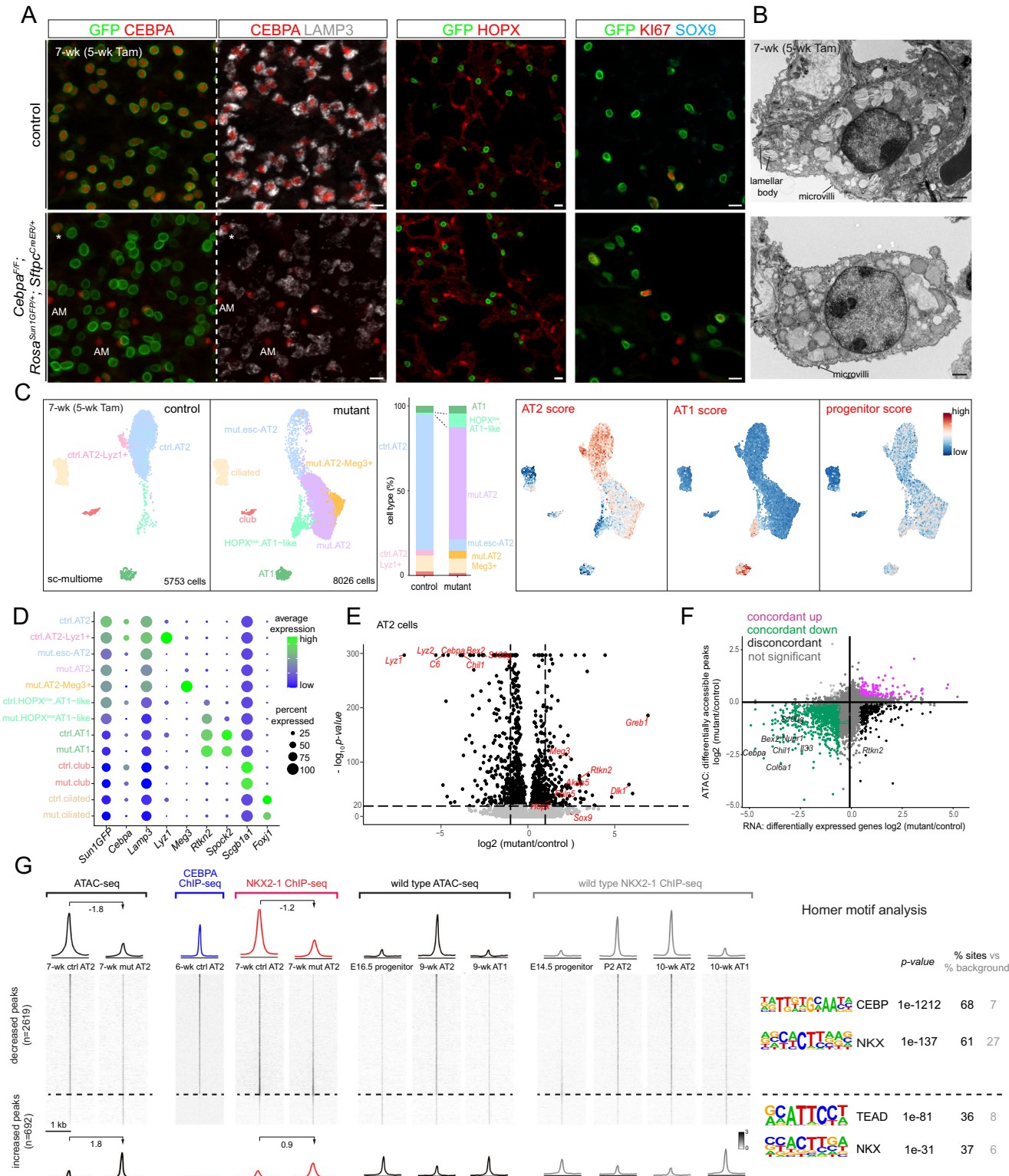

CEBPA binding, were enriched for NKX and TEAD motifs, but not SOX motif, and had some accessibility enriched for progenitors and AT1 cells but to a much lesser extent than the neonatal increased peaks (Fig. 5G, Supplementary Fig. 6D). NKX2-1 binding in purified control AT2 cells was low but increased in *Cebpa* mutant AT2 cells, possibly due to its limited redistribution to AT1-specific sites in the considerable number of HOPX[low] AT1-like cells (Fig. 5G).

The main difference between the mature versus neonatal *Cebpa* deletion models was the inability of mature AT2 cells to reactivate the

SOX9 progenitor program. This decrease in cell plasticity as AT2 cells matured were molecularly defined as the CEBPA-dependent, increased peaks unique to neonatal AT2 cells (5124 peaks in Supplementary Fig. 6E). The neonatal-specific plasticity was in regions that were accessible in progenitors and closed for 2 days versus 35 days when Cre-recombination was induced in neonatal versus mature lungs, respectively (Supplementary Fig. 6E). The duration of chromatin closure might lead to less reversible changes in histone modifications, DNA methylation, or high-order chromatin structure across the sites of

**Fig. 5 | CEBPA maintains the AT2 program without affecting the progenitor program in mature AT2 cells. A** Confocal images of immunostained adult AT2-specific *Cebpa* mutant and littermate control lungs showing loss of CEBPA in GFP+ recombined cells (asterisk: escaper), without affecting its expression in alveolar macrophages in the airspace (AM), and reduced LAMP3, but no extra HOPX, SOX9, or KI67. Tam, two doses of 3 mg each tamoxifen at 48 h interval (same for the rest of Fig. 5) (n = 3 mice each). Scale: 10 μm. **B** TEM images showing a reduction in lamellar bodies in mutant AT2 cells without affecting their apical microvilli. Large granules in mutant AT2 cells lack characteristic lamellae (n = 2 mice each). Scale: 1 μm. See Supplementary Fig. 6C for quantification. **C** Sc-multiome UMAPs of purified epithelial cells from *Cebpa* mutant and littermate control lungs color-coded by cell type (left), the corresponding percentages (middle), and metagene scores. Esc, escaper. See Source Data for metagene lists. **D** Dot plot showing the lineage marker (Sun1GFP), *Cebpa*, and cell type markers. *Rtkn2*, but not *Spock2*, is expressed in HOPX^low AT1-like cells. **E** Volcano plot (two-tailed, non-parametric Wilcoxon rank

sum test) showing downregulation of AT2 genes but minimal upregulation of progenitor/AT1 genes in mutant AT2 cells (left) compared to control AT2 cells (right) defined in (**C**). Compare with Fig. 3D. **F** Scatter plot correlating changes in the accessibility of scATAC-seq peaks (y-axis) and scRNA-seq expression of their nearest genes (x-axis), color-coded as concordant or discordant as well as the directionality of change. Compared to Fig. 3E, few concordant pairs are upregulated. See Source Data for the complete list. **G** Heatmaps and profile plots of decreased and increased scATAC-seq peak sets in the adult mutant and associated log2 fold changes, as well as the corresponding CEBPA and NKX2-1 binding and scATAC-seq data in wild type cells and associated Homer motifs. Decreased peaks have CEBPA binding and decreased NKX2-1 binding, corresponding to ATAC accessibility and NKX2-1 binding in wild type AT2 cells. Increased peaks are many fewer and have no CEBPA binding and increased NKX2-1 binding, corresponding to NKX2-1 binding in wild type AT1 cells.

differential plasticity or a few nodal sites of master genes, such as *Sox9*. Although CEBPA did not bind to these differentially plastic sites, its deletion revealed their presence.

To further define the temporal window of AT2 cell plasticity and potential regulators, we deleted *Cebpa* at additional neonatal time points and found robust SOX9 activation upon deletion at P4, but weak SOX9 at P7 and little SOX9 at P10 (Supplementary Fig. 6F). Accordingly, comparison of P4 and P10 scRNA-seq of AT2 cells revealed downregulated and upregulated genes that might promote and suppress plasticity, respectively (Supplementary Fig. 6G). Intriguingly and worthy of future investigation, *Dlk1* was among the most downregulated at P10 and had been implicated in antagonizing Notch signaling and AT2 self-renewal during injury repair[20].

Taken together, in neonatal and mature AT2 cells, CEBPA recruits NKX2-1 to promote and maintain the AT2 program; without CEBPA, neonatal but not mature AT2 cells have the plasticity to reactivate the SOX9 progenitor program.

## Viral infection expands CEBPA-dependent plasticity in mature AT2 cells

The temporal restriction in cell plasticity from neonatal to mature AT2 cells reminded us of the doctrine that injury-repair recapitulates development and prompted us to test if respiratory virus infection would reactivate the neonatal plasticity in mature AT2 cells. We infected our mature *Cebpa* deletion model with Sendai virus, which was known to preferentially injure AT2 cells, forming AT2-less regions, and trigger AT2 cell proliferation 14 days post infection[21]. Strikingly, while the infected control lung repaired itself with no SOX9 expression and only isolated KI67 expression in AT2 cells, the infected *Cebpa* mutant lung had SOX9 expression in 8% of AT2 cells (7776 GFP+ cells from 3 mice) and clusters of KI67 + AT2 cells, reminiscent of the neonatal *Cebpa* mutant (Fig. 6A). SOX9 + AT2 cells often abutted lobe edges or airways and macro-vessels, topologically distal ends of the respiratory tree favoring de novo growth as we described[21] (Supplementary Fig. 8A). The regional preference, in conjunction with localized virus delivery, suggested that the percentage of mutant AT2 cells capable of expressing SOX9 could be much higher. SOX9 activation depended on infection because saline treated control and *Cebpa* mutant lungs did not express SOX9 (Supplementary Fig. 8B).

E-Cadherin+ epithelial cells from infected control and *Cebpa* mutant lungs were profiled with single-cell multiome (Fig. 6E, F). As in prior neonatal and mature mutant models, *Cebpa*- AT2 cells from the infected mutant lung clustered separately from escapers of deletion, as well as AT2 cells in the infected control lung. Progenitor score and genes including *Sox9*, *Dlk1*, and *Kif4* were higher in the mutant, despite the said spatial restriction (Fig. 6G, Supplementary Fig. 9C). Proliferative AT2 cell cluster was much more prominent in the mutant, corroborating the KI67 immunostaining, and expressed *Sox9*, suggesting a possible coupling between proliferation and SOX9 activation

(Fig. 6F, G). Supporting this, despite their low percentage, SOX9 + AT2 cells were more likely to be KI67+ than SOX9- AT2 cells (45% vs 3.5%; 696 SOX9+ cells out of 8562 GFP+ cells from 3 mice; Fig. 6D). By comparison, KI67 + AT2 cells in the infected control lung were equally likely to be CEBPA+ or CEBPA- (5.9% vs 6.0%; 233 CEBPA- cells out of 2642 GFP+ cells from 3 mice; Fig. 6D), suggesting that transition of control AT2 cells into other CEBPA- populations, as examined further below, was uncoupled from proliferation. Despite the transcriptional resemblance, *Cebpa* mutant AT2 cells were not bona fide SOX9 progenitors since without CEBPA, they were not expected to meet the functional definition of differentiation to AT2 cells and they existed in an adult, injured niche different from their native embryonic environment.

Sendai virus infection also induced in both control and mutant lungs two other GFP + AT2-derived cell populations: KRT8/CLDN4+ transitional cells and AT1-like cells, marked by respective gene signatures (Fig. 6G). By immunostaining, the former had high KRT8 and ectopic CLDN4, but low LAMP3 and no HOPX; the latter had HOPX but no LAMP3 and were no longer cuboidal (Supplementary Fig. 8C, D). Although the two populations could represent sequential steps during AT2 to AT1 differentiation[22–25], their locations on the UMAPs were also compatible with two parallel states with only the AT1-like cells transitioning to AT1 cells and KRT8/CLDN4+ cells being arrested. Regardless, the *Cebpa* mutant lungs had a dramatic expansion of KRT8/CLDN4+ transitional cells, as confirmed by immunostaining, which were distinct from SOX9-expressing cells (Fig. 6B, Supplementary Fig. 9A). Despite the higher number of AT1-like cells captured by single-cell multiome, they were not reliably detected by HOPX immunostaining, possibly due to the higher sensitivity of single-cell multiome in documenting the gradual AT2-AT1 transition (Fig. 6C, D, E). CEBPA was normally lost in both populations even in the control lung by single-cell profiling and immunostaining (Supplementary Fig. 9B, C), consistent with their decreased/lost LAMP3 expression and the described role of CEBPA in maintaining the AT2 program. Therefore, the population expansion in the mutant was likely because most AT2 cells became eligible for alternative fates as the result of losing CEBPA. Furthermore, as the loss of CEBPA in KRT8/CLDN4+ and AT1-like cells alone was insufficient to activate *Sox9* in the infected control lung, SOX9 activation in the infected mutant represented a separate plasticity from injury-induced loss of the AT2 program and adoption of KRT8/CLDN4+ and AT1-like programs. Taken together, Sendai virus infection increases the plasticity of mature AT2 cells, which manifests upon *Cebpa* deletion as activation of the SOX9 progenitor program and expansion of the KRT8/CLDN4+ program.

## Discussion

In this study, we have tracked the transcriptomic and epigenomic changes as well as plasticity of AT2 cells during their specification and subsequent maturation and upon viral injury. We show that the AT2-

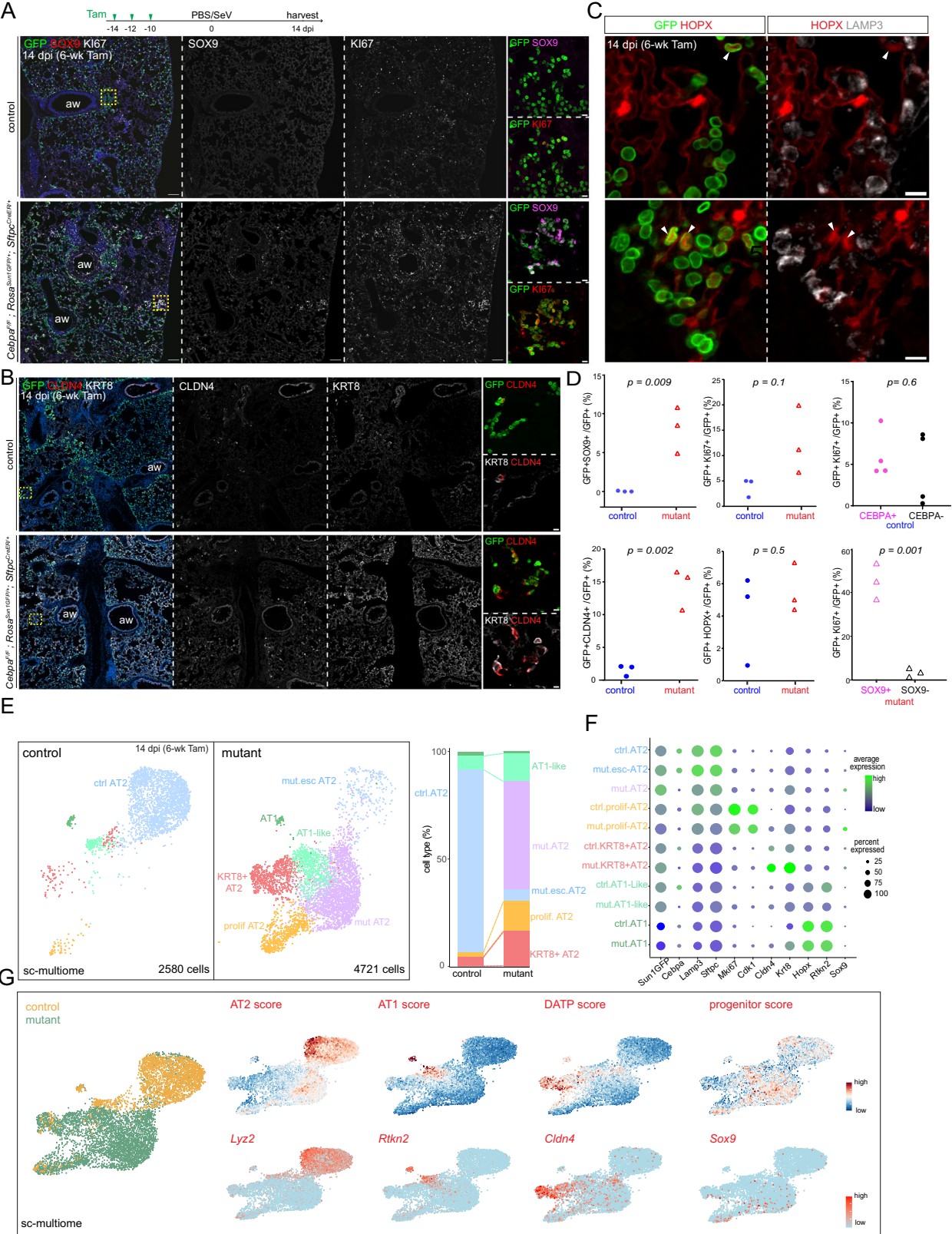

specific TF CEBPA recruits the lung lineage TF NKX2-1 to promote and maintain the AT2 program. *Cebpa* deletion also reveals an evolving landscape of AT2 cell plasticity shaped by the developmental history and external stimuli. In wild type lungs, neonatal and mature AT2 cells form AT1-like cells infrequently during homeostasis, but more readily upon viral injury and additionally form KRT8/CLDN4+ transitional cells

(Fig. 7A). Without CEBPA, these processes are enhanced; most strikingly, neonatal but not mature AT2 cells activate the SOX9 progenitor program and this neonatal plasticity is bestowed to mature AT2 cells by viral injury (Fig. 7A). Rather than direct binding and regulation by CEBPA, the plasticity landscape is submerged by the AT2 program, partially exposed when altered sufficiently, but unmasked upon *Cebpa*

**Fig. 6 | Viral infection expands CEBPA-dependent plasticity in mature AT2 cells.**
**Figure 6. Viral infection expands CEBPA-dependent plasticity in mature AT2 cells. A** Experimental timeline of tamoxifen injection (Tam, 3 mg), Sendai virus (SeV) or saline (PBS) administration, and lung harvest at 14 dpi (day post-infection). Confocal images of immunostained infected Cebpa mutant and littermate control lungs, showing mutant-specific activation of SOX9 and increase in KI67 near airways (aw) and lobe edges (inset; scale: 10 μm). Scale: 100 μm. **B** Confocal images of lungs in (**A**) showing increased KRT8 and CLDN4. Scale: 100 μm (inset: 10 μm). **C** Confocal images of lungs in (**A**) showing lineage-labeled HOPX+ cells with little LAMP3 (arrowhead). Scale: 10 μm. **D** Quantification of (**A**, **B**, **C**). KI67+ cells in the

control and Cebpa mutant are stratified by CEBPA and SOX9 expression, respectively. Each symbol represents one mouse from littermate pairs. *P* values were calculated using two-tailed Student's *t* test. **E** Sc-multiome UMAPs of purified epithelial cells from infected Cebpa mutant and littermate control lungs color-coded by cell type (left) and the corresponding percentages (right). Esc, escaper; prolif, proliferating. **F** Dot plot showing the lineage marker (Sun1GFP), Cebpa, and cell type markers. **G** Sc-multiome UMAP color-coded for genotype (left) and feature plots of metagene scores (top) and representative genes (bottom). A published damage-associated transient progenitor (DATP) score marks KRT8/CLDN4+ cells.

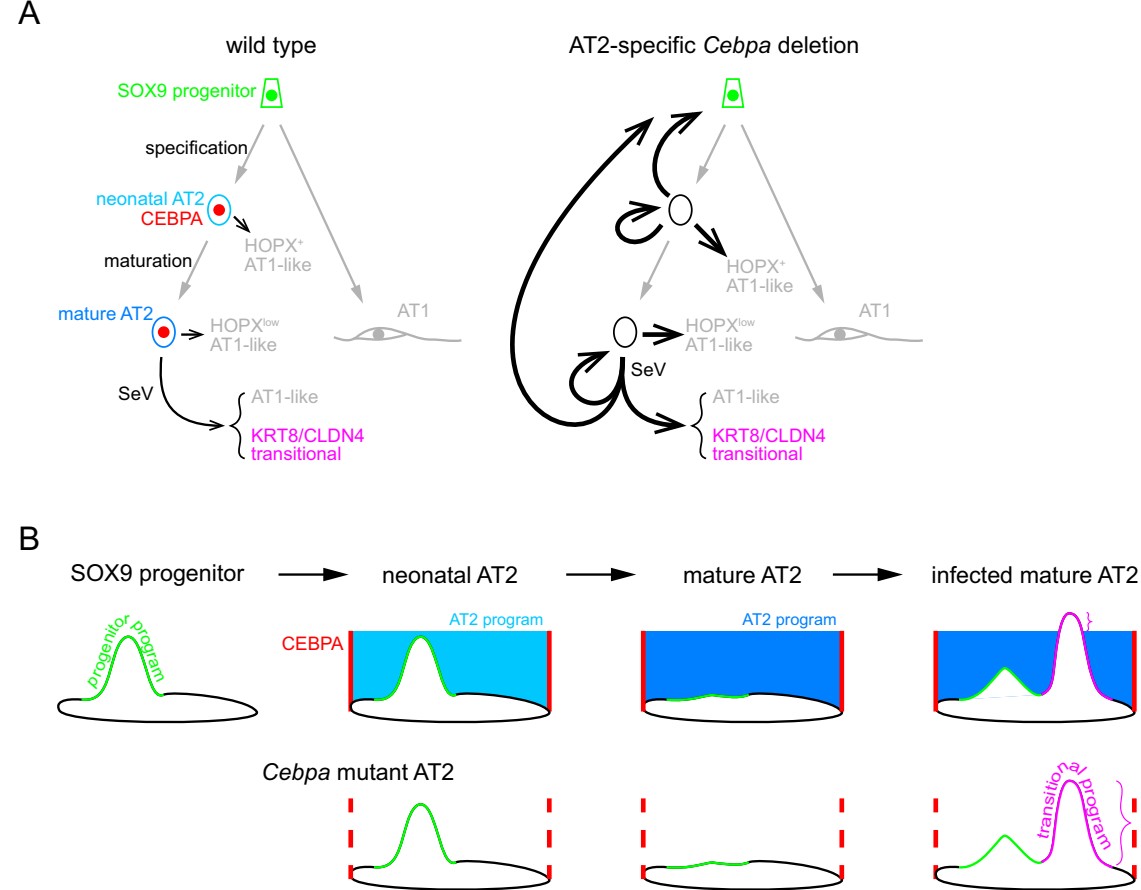

**Fig. 7 | Diagram of CEBPA restricting AT2 cell plasticity during development and injury-repair. A** In a wild type lung, SOX9 progenitors undergo specification and maturation to become neonatal/nascent and mature AT2 cells, sequentially, expressing CEBPA. A small fraction of AT2 cells at both stages become AT1-like cells with varying levels of HOPX. Sendai virus (SeV) infection induces KRT8/CLDN4 transitional cells as well as AT1-like cells, as a sequential or parallel response of injury-repair. *Cebpa* deletion in neonatal, but not mature, AT2 cells leads to reversion to the progenitor program and proliferation, in addition to an increase of AT1-like cells upon either deletion. Sendai virus infection bestows mature AT2 cells with the plasticity to revert to progenitors as well as to more readily transition to

the KRT8/CLDN4 state. **B** In an aerial view of the cell plasticity landscape, the progenitor program, although still present in neonatal AT2 cells, is submerged under the AT2 program promoted by CEBPA. Although CEBPA does not bind to genomic regions of plasticity, without CEBPA, the AT2 program subcedes to expose the progenitor program. As AT2 cells mature, the progenitor program disappears so that, even without the AT2 program, no progenitor program is visible. After infection, mature AT2 cells reshape their plasticity landscape, but only the KRT8/CLDN4 transitional program rises enough to manifest over the AT2 program. Without CEBPA and the AT2 program, the progenitor program is visible and the transitional program more evident.

deletion (Fig. 7B). This interplay between the sea-level program and submerged plasticity landscape could be important in generating and engrafting the most effective lung cells in cell therapy and in activating endogenous stem cells without forming a tumorigenic landscape.

A gene regulatory network of alveolar cell fates is emerging from recent mechanistic studies of alveolar TFs[8,26]. Both AT1 and AT2 cells express the lung lineage TF NKX2-1 and require NKX2-1 to express AT1 or AT2-specific genes, respectively, and to suppress gastrointestinal (GI) genes. The cell-type-specific functions of NKX2-1 are attributed to its cell-type-specific chromatin binding, as the result of its recruitment

by cell-type-specific co-TFs: YAP/TAZ/TEAD in AT1 cells and CEBPA in AT2 cells[8]. This symmetry between AT1 and AT2 cells is skewed with respect to cell plasticity. Neonatal *Yap/Taz* mutant AT1 cells readily become AT2-like, whereas only 12% neonatal *Cebpa* mutant AT2 cells become AT1-like with the vast majority (80%) activating the SOX9 progenitor program (Fig. 2). Perhaps AT2 cells are more closely related to progenitors in gene expression and cell morphology than AT1 cells are, as possibly needed for their facultative stem cell function. Alternatively but compatibly, AT2 differentiation might be the default path of SOX9 progenitors whereas AT1 differentiation requires YAP/TAZ

activation by mechanical stretching in the growing neonatal lung. As a result, *Yap/Taz* mutant AT1 cells fail to relay the mechanical signal to AT1-specific NKX2-1 recruitment and default to AT2 differentiation, whereas *Cebpa* mutant AT2 cells, surrounded by normal AT1 cells, are not subject to mechanical stretching and AT1 differentiation. The neonatal plasticity becomes limited in the mature lung where *Yap/Taz* mutant AT1 cells less readily become AT2-like and *Cebpa* mutant AT2 cells do not become SOX9 progenitors[27]. Sendai virus injury reactivates the SOX9 progenitor program in *Cebpa* mutant AT2 cells, whereas hyperoxia injury or blocking mechanical stretching stimulates the AT2 program even in wild type AT1 cells, although more studies are needed to understand the predicted shedding of excessive cell membrane during AT1-to-AT2 conversion[27,28]. Future studies are expected to expand the core NKX2-1/YAP/TAZ/TEAD/CEBPA network to include additional TFs, such as KLF5 and AP-1 members, and effector enzymes, such as histone modifiers and chromatin remodelers[29], providing a genome-wide mechanistic understanding of regulators of lung development from genetic studies[30].

This study illustrates an experimental and conceptual framework of cell plasticity and highlights three associated challenges: its diversity, definition, and regulation. First, cell plasticity can be forced through overexpression of TFs during cell reprogramming in culture or in vivo[1,31]. Although cell plasticity from gene deletion is likely more physiological, both deleting and overexpressing different TFs in the same cell can lead to different plasticity, as seen in the AT1-AT2-progenitor plasticity triad from targeting YAP/TAZ/CEBPA versus the lung-GI dyad from targeting NKX2-1 (this study and[8,26]). Instead of submitting to the notion of universal plasticity from arbitrary use of the same DNA blueprint, we need systematic precise gene perturbation coupled with quantitative molecular readouts of cell plasticity.

Relatedly, the second challenge arises from qualitative, categorical, or at times semantic definitions of a cell type. Genome-wide profilings have revealed molecular variants of classical cell types, invoking prefixes such as pro-, pre-, semi-, and quasi- and postfixes such as -like and -oid, as well as adjectives such as intermediate, transitional, and hybrid – several of which are also used in this study. As experimental and computational technology advances, cell types and states will be objectively digitized and their changes upon perturbations become the definition of cell plasticity. Our use of the accessibility changes upon *Cebpa* deletion in neonatal or mature AT2 cells is such an attempt. A better definition of cell type and cell plasticity will also help clarify comparable terms in the literature including cell potency, lineage infidelity, and points of no return[32–34].

Last, although we show that CEBPA restricts and its deletion reveals AT2 cell plasticity, the molecular correlates, and regulators of the shifting plasticity landscape are unclear. Sites of plasticity may be primed/poised via bivalent histone marking, opened de novo by pioneering TFs, or reset upon DNA/histone synthesis during cell cycle[35–37]. Regardless, CEBPA does not bind to plasticity sites, as in the lack of binding of NKX2-1 to GI genes[26]. The recruitment model predicts free diffusion and ectopic binding of the partner TF in the absence of the recruiting TF, such that FOXA2 might activate GI genes without NKX2-1 and NKX2-1 might activate AT2 genes without YAP/TAZ[8,38]. However, there is limited gain of NKX2-1 binding at the plasticity sites in both neonatal and mature *Cebpa* mutant AT2 cells (Figs. 4, 5). Rather than redistributing CEBPA's partner TF NKX2-1, the neonatal *Cebpa* mutant cells transcriptionally activate SOX9, whose motif causally or coincidentally dominates the plasticity sites. These plasticity sites, such as the one near *Sox9*, are open in SOX9 progenitors several days prior and become closed and unresponsive to *Cebpa* deletion in mature AT2 cells, suggesting a role of developmental history in shaping cell plasticity. Reactivation of developmental plasticity upon viral injury (Fig. 6), or possibly during tumorigenesis, may rewrite cells' developmental history and represent therapeutic opportunities.

## Methods

### Mice
*Cebpa^{F/F}*[39] were obtained from the Jackson Laboratory (stock #006447). *Sftpc^{CreER}*[40] and *Rosa^{Sun1GFP}*[41] have been previously described. Mice were given intraperitoneal injections of tamoxifen (T5649, Sigma) dissolved in corn oil (C8267, Sigma). The doses and time of injection were described in figure legends. The animals were housed at 22 °C 45% humidity, and 12-12-h light-dark cycle conditions. The mice used in the experiments were of mixed genders and were of C57BL/6 and 129 mixed backgrounds. The age and the number of mice were described in the figure legend. To minimize experimental variability, samples were processed using the same tissue blocks or tubes. The sample sizes were not calculated based on power analysis. All mice were housed in the MD Anderson facility and the proposed studies was performed following all federal regulations on the use of animals in research and have been approved by the Institutional Animal Care and Use Committee at the University of Texas MD Anderson Cancer Center.

### Antibodies
For immunofluorescence, the following antibodies were used: rabbit anti-CCAAT/enhancer binding protein alpha (C/EBPA, 1:500, 8178 P, Cell Signaling Technology), chicken anti-green fluorescent protein (GFP, 1:5000, AB13970, Abcam), rabbit anti-NK homeobox 2-1 (NKX2-1, 1:1000, sc-13040, Santa Cruz), rabbit anti-pro-surfactant protein C (SFTPC, 1:1000, AB3786, Millipore), goat anti-SOX9 (SOX9, 1:1000, AF3075, R&D Systems), rabbit ani-SOX9 (SOX9, 1:1000, AB5535, Millipore). Goat Anti-Mouse Il-33 (IL33, 1:500, R&D, AF3626). rabbit anti-homeodomain only protein (HOPX, 1:500, sc-30216, Santa Cruz). mouse anti-homeodomain only protein (HOPX, 1:250, sc-398703 AF647, Santa Cruz). rat anti-KI67 (KI67, 1:1000, 14-5698-82, Invitrogen), guinea pig anti-lysosomal associated membrane protein 3 (LAMP3, 1:500, 391005, SySy), rat anti-epithelial cadherin (ECAD, 1:1000, 13190, Invitrogen).

The following antibodies were used for FACS: PE/Cy7 rat anti-CD45 (CD45, 1:250, 103114, BioLegend), PE rat anti-epithelial cadherin (ECAD, 1:250, 147304, BioLegend), BV421 rat anti-epithelial cell adhesion molecule (EPCAM, 1:250, 118225, BioLegend), and AF647 rat anti-Intercellular adhesion molecule 2 (ICAM2, 1:250, A15452, Thermo Fisher).

The following antibodies were used for chromatin immunoprecipitation: rabbit anti-NK Homeobox 2-1 (NKX2-1, 1 μg per reaction, ab133737, Abcam) and rabbit anti-CEBPA (C/EBPα, (D56F10) XP, 1 μg per reaction, Cell Signaling Technology).

### Section immunofluorescence
After intraperitoneal injection of Avertin (Sigma, T48402), the heart's right ventricle was subjected to perfusion using phosphate-buffered saline (PBS, pH 7.4). Subsequently, the trachea was cannulated, and the lungs were inflated with a solution of 0.5% paraformaldehyde (Sigma, P6148) in PBS, maintaining a pressure of 25 cm H₂O. The lungs were fixed in 0.5% PFA in PBS for 3–4 h at room temperature and then washed with PBS overnight at 4 °C. For section immunostaining, the fixed lung lobes were cryoprotected overnight at 4 °C in 20% sucrose in PBS containing 10% optimal cutting temperature compound (OCT; 4583, Tissue-Tek, Tokyo, Japan; optional for embryonic lungs) and then embedded in OCT and frozen in −80 °C. Immunostaining of frozen sections was adapted from previously described method[42]. Briefly, frozen sections at 10-μm thickness were blocked in PBS with 0.3% Triton X-100 and 5% normal donkey serum (017-000-121, Jackson ImmunoResearch) and then incubated with primary antibodies diluted in PBS with 0.3% Triton X-100 at 4 °C overnight in a humidified chamber. The following day, the sections were washed with PBS for 1 h at room temperature and then incubated with secondary antibodies (Jackson ImmunoResearch) and 4′,6-diamidino-2-phenylindole (DAPI)

diluted in PBS with 0.1% Triton X-100 and 0.1% Tween-20 for 90 min at room temperature. The sections were then washed as described above and mounted with Aquamount mounting medium (18,606, Polysciences) and imaged using Olympus FV1000 confocal microscope, and quantified with the Imaris software (version 7.7.2).

## Whole mount immunostaining

For whole mount immunostaining, previously published protocol was followed with minor modifications[43]. Briefly, ~3 mm wide strips were cut from the periphery of cranial or left lobes. Strips were blocked in PBS with 0.3% Triton X-100 and 5% normal donkey serum and then incubated with primary antibodies diluted in PBS with 0.3% Triton X-100 at 4 °C overnight in an 1.7 ml tube. The next day, the strips were washed with PBS + 1% Triton X-100 + 1% Tween-20 (PBSTT) on a rocker at room temperature for 1 h and this wash was repeated 3 times. Secondary antibodies were diluted in PBS with 0.3% Triton X-100 and added to the strips for an overnight incubation on a rocker at 4 °C. The third day, strips were washed was PBSTT as described and then fixed with 2% PFA in PBS for 3 h. Strips were washed 3 times with PBS and mounted on slides using Aquamount (18606, Polysciences) with the flat side of the strips facing the coverslip. Z-stacks of 20–30 μm thickness at 1 μm step size were taken using Olympus FV1000 confocal microscope and quantified with the Imaris software (version 7.7.2).

## Transmission electron microscopy

Samples were fixed with a solution containing 3% glutaraldehyde plus 2% paraformaldehyde in 0.1 M cacodylate buffer, pH 7.3, then washed in 0.1 M sodium cacodylate buffer, treated with 0.1% Millipore-filtered cacodylate buffered tannic acid, post fixed with 1% buffered osmium tetroxide, and stained en bloc with 1% Millipore-filtered uranyl acetate. The samples were dehydrated in increasing concentrations of ethanol, infiltrated, and embedded in LX-112 medium. The samples were polymerized in a 60 °C oven for approximately 3 days. Ultrathin sections were cut in a Leica Ultracut microtome (Leica, Deerfield, IL), stained with uranyl acetate and lead citrate, and examined in a JEM 1010 transmission electron microscope (JEOL, USA, Inc., Peabody, MA) at an accelerating voltage of 80 kV. Digital images were obtained using AMT Imaging System (Advanced Microscopy Techniques Corp, Danvers, MA).

## Sendai virus infection

Viral infection was carried out following the previously described procedure[21]. Briefly, mice anesthetized with isoflurane were held by their upper incisors and subjected to oropharyngeal instillation of a non-lethal dose of Sendai Virus (ATCC #VR-105, RRID:SCR_001672CSCSSCS), approximately $2.1 \times 10^7$ plaque-forming units, suspended in 40 μl of PBS. The control group received 40 μl of PBS.

## Tissue dissociation and fluorescence-activated cell sorting

Mouse lungs were collected as described above and subjected to our published protocol with minor modifications[26]. Briefly, connective tissues and trachea were removed, and lungs were minced to small pieces using forceps. The lungs were digested at 37 °C for 30 min in 1.35 ml Liebovitz media (Gibco, 21083-027) with the following enzymes: 2 mg/ml collagenase type I (Worthington, CLS-1, LS004197), 0.5 mg/ml DNase I (Worthington, D, LS002007), and 2 mg/ml elastase (Worthingon, ESL, LS002294). To stop the enzymatic reaction, 300 μl fetal bovine serum (FBS, Invitrogen, 10082-139) were added to a final concentration of 20%. The digested tissue was mixed at least 10 times with pipette and filtered through a 70 μm cell strainer (Falcon, 352350) on ice in a 4 °C cold room and transferred to a 2 ml tube. The cells were pelleted at 1537 g for 1 min. Then the supernatant was removed and 1 ml of red blood cell lysis buffer (15 mM $NH_4Cl$, 12 mM $NaHCO_3$, 0.1 mM EDTA, pH 8.0) was added and samples incubated on ice for 3 min. Cells were pelleted again by centrifugation at 1537 g for 1 min, washed with once Liebovitz + 10% FBS, resuspended with 1 ml Liebovitz + 10% FBS and filtered through

35 μm cell strainer into a 5 ml glass tube. Samples for scMultiome were stained with CD45-PE/Cy7 (BioLegend, 103114), ECAD-PE (BioLegend, 147304), and ICAM2-A647 (Invitrogen, A15452) antibodies (1:250 dilutions for all antibodies) as well as SYTOX Blue (1:1000, Invitrogen, S34857) for 30 min on ice. Then samples were washed and resuspended with Liebovitz + 10% FBS, filtered again through a 35 μm strainer into a 5 ml glass tube, and sorted using Aria II Cell sorter with a 70 μm nozzle. Cell sorting data were analyzed using FlowJo 10.7. Lung epithelial cells from littermate $Cebpa^{F/F}$; $Rosa^{Sun1GFP/+}$; $Sftpc^{CreER/+}$ ($n = 2$ mice each) and $Cebpa^{F/+}$; $Rosa^{Sun1GFP/+}$; $Sftpc^{CreER/+}$ lungs ($n = 2$ mice each) were purified with a cell viability >77%. Single cell libraries were prepared using the Single Cell Multiome ATAC + Gene Expression kit (10x Genomics) and 10,000 nuclei were loaded per lane.

## Nuclei sorting for cell-type specific ChIP-seq

Lungs were collected as described above. Nuclei isolation followed the previously published protocol[8]. Briefly, lungs were minced with forceps and transferred to a 5 ml glass tube and crosslinked with 2 ml of 1% formaldehyde diluted with PBS from 10% buffered formalin (i.e., 3.7% formaldehyde;Thermo Fisher Scientific, 23-245-685) for 10 min at room temperature on a rocker. The excess formaldehyde was quenched by adding 0.5 M glycine (pH 5.0) to 0.125 mM final concentration and incubated for 15 min at RT on a rocker. The tissue was washed twice with ice cold PBS and resuspended and homogenized with Douncer homogenizer for 5 strokes in 1 ml of Isolation of Nuclei Tagged in specific Cell Types (INTACT) buffer (20 mM HEPES pH 7.4, 25 mM KCl, 0.5 mM $MgCl_2$, 0.25 M sucrose, 1 mM DTT, 0.4% NP-040, 0.5 mM Spermine, 0.5 mM Spermidine)[41] with protease inhibitor cocktail (cOmplete ULTRA Tablets, Mini, EDTA-free, EASY pack, Sigma, 5892791001). The tissue was then filtered through a 70 μm strainer and then transferred to a 2 ml tube coated with 10 mg/ml BSA (Sigma, A3059). The nuclei were centrifuged at 384 g for 5 min 4 °C, resuspended with 1 ml ice cold PBS plus protease inhibitor cocktail, and then filtered through a 35 μm strainer into a 5 ml glass tube coated with 10 mg/ml BSA. SYTOX Blue was added with at a 1:1000 dilution. SYTOX + and GFP+ nuclei were collected on Aria II Cell sorter with a 70 μm nozzle at 4 °C into a 5 ml glass tube coated with 10 mg/ml BSA and containing 300 μl of 10 mg/ml BSA with 5x protease inhibitor cocktail in PBS. $Sftpc^{CreER/+}$; $Rosa^{Sun1GFP/+}$ mice yielded 1–2 million GFP+ nuclei per adult lung. Cell sorting was analyzed using FlowJo (version10).

## Chromatin immunoprecipitation

**Lysis of the nuclei and shearing of the chromatin.** AT2-specific chromatin immunoprecipitation (ChIP) was performed on sorted nuclei following the previously published protocol with minor modification[8]. Briefly, sorted nuclei were split to aliquots of 1 million in 1.7 ml tubes and pelleted by centrifugation at 6708 g for 10 min at 4 °C. The supernatant was discarded and pellets were resuspended in 100 μl of nuclei lysis buffer plus proteinase inhibitor and incubated on ice for 15 min. Samples were sonicated using Diagenode Bioruptor (B01060010) precooled to 4 °C for 36 cycles of 30 s ON/30 s OFF to achieve a DNA fragment size of ~200–500 bp. Samples were then centrifuged at 13,148 g for 10 min at 4 °C and the supernatant was transferred to new 1.7 ml tubes. 20 μl of samples were saved as the input control in a separate tube and stored in −20 °C till the reverse crosslinking step. Bead Preparation. Two sets of Protein G Dynabeads (Thermo Fisher Scientific, 10004D) were washed twice with 1 ml ChIP dilution buffer (16.7 mM Tris-HCl pH 8.1, 1.2 mM EDTA, 1.1% Triton X-100, 0.01% SDS with 1× protease inhibitor cocktail) and then blocked with 200 μl 20 mg/ml BSA (Jackson ImmunoResearch, 001-000-161), 4 μl 10 mg/ml salmon sperm DNA (Invitrogen, 15632-011) in the ChIP dilution buffer. The first set of Protein G Dynabeads was blocked for 1 h on a rotator at 4 °C and was used to preclear the chromatin (40 μl of protein G Dynabeads per sample). The second set was blocked overnight (40 μl of protein G Dynabeads per sample) and was used for immunoprecipitation the second day. Preclearing and antibody

incubation. After chromatin shearing and centrifugation, samples were diluted to 1 ml with ChIP dilution buffer. The first set of beads were washed twice with ChIP dilution buffer using a magnetic adaptor before adding to samples, which were then incubated on a rotator at 4 °C for 1 h. Using a magnetic adaptor, precleared samples were transferred to a new 1.7 tube and incubated with an adequate amount of antibody overnight at 4 °C on a rotator. Immunoprecipitation. The next day, the second set of beads were washed twice with ChIP dilution buffer and added to the samples (chromatin/Ab solution) and incubated for 3 h on a rotator at 4 °C. The sample were sequentially washed with 1 ml of the following prechilled buffers: low salt buffer (150 mM NaCl, 2 mM EDTA, 1% Triton X-100, 20 mM Tris-HCl pH 8.1, 0.1% SDS), high salt buffer (500 mM NaCl, 2 mM EDTA, 1% Triton X-100, 20 mM Tris-HCl pH 8.1, 0.1% SDS), lithium chloride buffer (250 mM LiCl, 1 mM EDTA, 1% NP-40, 10 mM Tris-HCl pH 8.1, 0.1% sodium deoxycholate), and TE buffer (10 mM pH 8.0 Tris and 1 mM EDTA) twice. Samples were resuspended in 300 μl TE buffer. Reversal of crosslinking and DNA purification. The frozen inputs were thawed and diluted to 300 μl with TE buffer. Then samples and inputs were incubated for 4 h at 37 °C with 1.5 μl of 10 mg/ml RNase A (Qiagen, 1007885) and 15 μl 10% sodium dodecyl sulfate and 3.5 μl 20 mg/ml Proteinase K (Thermo Fisher, EO0491). Then samples were switched to 65 °C overnight incubation. The next day, 300 μl of phenol: chloroform: isoamyl alcohol solution (Sigma, P2069-400 ML) was added to samples and input and mixed by full-speed vortexing for ~2 s, then transferred to phase-lock tubes (Qiagen MaXtract, 129046) and centrifuged for 5 min at RT at 13,148 g. 300 μl of chloroform was added to samples and mixed by inversion then centrifuged again at 13,148 g for 5 min. The top liquid phase was transferred to a new 1.7 ml tube containing 2 μl of 20 μg/μl glycogen (Invitrogen, 10814-010). Then 600 μl of 100% ethanol and 30 μl of 3 M NaCl were added and mixed by quick vortexing, then samples were stored at −20 °C overnight. Samples were centrifuged at 13,148 g for 10 min at 4 °C, then supernatant was discarded. The pellets were rinsed with 1 ml of 100% ethanol and centrifuged at 13,148 g for 1 min at 4 °C. The liquid was discarded, and pellets were air dried for 5 min and dissolved in 8 μl nuclease-free H$_2$O. ChIP-seq library preparation. DNA quantity was measured using a Qubit dsDNA HS Assay Kit (Invitrogen, Q23851). Then <5 ng ChIP sample DNA or <20 ng input DNA was used for sequencing libraries using the NEB Next Ultra II DNA Library Prep Kit for Illumina (New England BioLabs, E7645). In Step 1.3 (End Prep), thermocycler condition was modified as following: the heated lid set to ≥60 °C, 30 min at 20 °C then 60 min at 50 °C; as decreasing the temperature to 50 °C helps saving small fragments. The DNA was PCR amplified for 12 cycles using indexed primers (New England BioLabs, E7335S or E7500S) to barcode samples. Size selection and purification was achieved by double-sided (0.65 × − 1× volume) using SPRIselect magnetic beads (Beckman Coulter, B23318). Concentrations were measured using the Qubit HS dsDNA assay. Samples were pooled with less than 20 barcoded samples per sequencing run on an Illumina NextSeq500.

## Bioinformatics analyses

**ScRNA-seq time course analysis.** Published scRNAseq dataset (GSE158192)[8] compiled from 12-time points E14.5, E16.5, E18.5, P4, P6, P7, P8, P10, P15, P20, 10-week-old, and 15-week-old lung was analyzed using Seurat (version 4.1). Cells were filtered out if they had a gene count of less than 200 or over 5000. Epithelial, immune, mesenchymal, and endothelial lineages were identified based on the expression of *Cdh1*, *Ptprc*, *Col3a1*, and *Icam2*, respectively. Doublets were filtered out and epithelial cells were subset and re-clustered. AT2 cells and SOX9 progenitors were subset from epithelial cells and used for downstream analysis. AT2-specific genes used in the heatmap were obtained by differential gene expression comparison between 15-week AT2 and AT1 cells using 'Findmarkers' resulting in 88 distinct genes. Subsequently, the expression level of these 88 genes was compared across different time points: E16.5, E18.5, and 15-wk. Genes were classified as

early genes if they were present in at least 25% of E16.5 progenitor cells. The remaining genes were classified as late genes. Within this group, genes were classified as "mature" if the logFC between the 15-week time point and E18.5 exceeded 2 (i.e., 4-fold increase) (Source Data). Out of the 88 gene list, 6 genes were excluded (*Tpt1, Mt1, Mt2, Selenop, Wfdc2 and Ly6e*) as they met both early and mature criteria resulting in 82 genes that were used to generate heatmap in Fig. 1. Monocle (version 2.22.0) was used to analyze SOX9 progenitors and AT2 cells using the top 2000 genes to generate pseudo-time trajectories. Heatmap was generated by pheatmap (version 1.0.12) and ggpubr (version 0.6.0).

**Pseudobulk ATAC-seq time course analysis.** ScATAC-seq control datasets were from 5 different time points (E16.5, E18.5, P3, P8, and 9-wk) (GSE264098). The P3 dataset was previously published[14] and obtained from (GSM4504962). Cell barcodes of AT2 or SOX9 progenitors were identified based on *Nkx2-1* and *Sftpc* or *Sox9* accessibility. Each sample was randomly divided cell barcodes into two to generate replicates. Then Sinto (version 0.4.0) was used to subset the *−F 2048 −q 30*. Peaks were called using MACS2 (version 2.1.2) commands: *'q 0.05 −nomodel −shift -100 −extsize 200 −broad'*. Peaks overlapping with the mm10 blacklist were removed using bedtools (version 2.30.0). All overlapping peaks (>1 kb) from different time points were merged using bedtools (version 2.30.0) to obtain a reference peak set which was used for counting raw reads for each time point using Rsubread (version 2.8.2) and differential analysis was carried out by DEseq2 (version 1.34.0). Regions with (log2FC > 1, FDR < 0.01) were kept for downstream analysis. Diffbind (version 3.4.11) was used to obtain differentially accessible peaks representing the 4 categories: early lost (E16.5 vs E18.5); late lost (E18.5 vs 9-wk); early gain (E18.5 vs E16.5); and late gain (9-wk vs E18.5). GO analysis for each ATAC-seq cluster using Genomic Regions Enrichment of Annotation Tool (GREAT, version 4.0.4)[44] using the default setting and the whole mouse genome (GRCm38/mm10) as the background.

**Analysis of ChIP-seq data.** Reads were concatenated and their quality was assessed by FastQC (version 0.11.8) (http://www.bioinformatics.babraham.ac.uk/ projects/fastqc/). Trimmomatic (version 0.33)[45] was used to filter poor-quality reads and trim poor-quality bases. Then reads were aligned to the mm10 reference genome using Bowtie2 (version 2.4.1) with the following parameters: -m1 -k1 -v1. Samtools (version 1.15) was used to convert aligned sam files to bam files. Then files were deduplicated and filtered for unmapped reads and low-quality alignments using Picard's (version 2.9.0) MarkDuplicates and Samtools (version 1.15) settings: '-b -h -F 4 -F 1024 -F 2048 -q 30'. Peaks were called using MACS2 (version 2.4.1) with NarrowPeak setting '-g mm -n -B' for NKX2-1 and CEBPA. Peaks then were filtered for sites overlapping with the mm10 blacklist. Differential binding for NKX2-1 and CEBPA was carried out with Diffbind (version 3.4.11) normalized for sample read depth and at a fixed peak width of 500 bp between controls and mutants. Bigwig files were generated using bamCoverage (version 3.3.2) with the following setting: '−binSize 20 −normalizeUsing BPM −smoothLength 60 −centerReads'. Heatmaps and genomic trackers were generated using EaSeq (version 1.2) (http://easeq.net)[46]. ChIP-Seq peaks were assigned to the nearest gene using ChIPseeker (version 1.3). CEBPA and NKX2-1 co-bound peaks were identified using ChIP-peakAnno (version 3.2) using findOverlapsOfPeaks with a max gap 50 bp between peaks. Two replicates for each antibody were intersected using bedtools (version 2.30.0) with the *-wa -wb* parameter.

## Motif analyses

De novo motif analysis was obtained using Homer (version4.10)[47] 'findMotifsGenome.pl -size 200 -mask' with random backgrounds, which does not provide corrected *P*-values. Histograms of motif densities were performed using Homer 'annotatePeaks'. To calculate the average distance between NKX and CEBP motifs, we used Homer's

'annotatePeaks.pl -size 200' to search for NKX and CEBP motifs in NKX2-1/CEBPA co-bound peaks and calculated the distance from peak centers. ChromVAR (version 1.16.0) was used to test overrepresented motifs in differentially accessible peaks by calculating motif scores on the cell level. ChromVar z-score deviation scores were calculated for the top 100 motifs curated from JASPAR (2020 Version) motif database.

### Single-cell multiome analysis

Control and mutant samples were aggregated using cellranger-arc (version 2.0.0) via "cellranger-arc count" and "cellranger-arc aggr" command using a custom mm10 reference genome that contains the *Sun1GFP* transcript. Downstream analysis was carried out in R (version 4.1.1) using the Seurat R package (version 4.3) for RNA and Signac (version 1.9) for ATAC and combined with the weighted nearest neighbor method. The signature score was calculated via the Seurat module score function using our previously published 119 progenitor genes, 100 AT2 genes, and 100 AT1 genes[48]. 89 transitional genes (DATPS score) were derived from Choi et al.[25]. Psudobulk ATAC profiles for AT2 cells were generated using Sinto as described. Mutant gained and lost peaks were obtained using Diffbind. Overlapped peaks were obtained using bedtools intersect with parameters '-wa -wb'. Unique peaks were obtained by bedtools subtract with parameters '-A'. To generate RNA-seq/ATAC-seq scatter plots, differentially accessible peaks between mutant and control AT2 cells excluding proliferative cluster were linked to genes via Signac's function LinkPeaks using the method described by SHARE-seq[49]. A custom script inspired by a prior study[50] was used to combine gene expression and chromatin accessibility.

### RNA velocity analysis

RNA velocity was conducted using python (version 3.11.5) to estimate the spliced and unspliced counts in our scMultiome data. We used velocyto (version 0.17.17) with the 'velocyto run10x' command and scVelo (version 0.3.1) (https://github.com/theislab/scvelo) for downstream analysis[51,52]. The 'scvelo.tl.velocity' (version 0.3.1) function was employed with the mode parameter set to 'stochastic' to calculate cell-specific RNA velocities. Subsequently, these velocities were projected onto the UMAP coordinates derived from weighted nearest neighbor analysis (wnn.UMAP) for visualization purposes, using the scvelo.tl.velocity_embedding() function.

All bioinformatics scripts are included in the Source Data file.

### Statistics and reproducibility

Cumulative binomial distribution was used to calculate the significance of Homer motif enrichment in Figs. 1D, 3F, and 5G. All confocal images are representative of at least three imaging fields of each sample and at least three sets of control and mutant lungs. Hundreds to thousands of cells were quantified in each comparison.

### Reporting summary

Further information on research design is available in the Nature Portfolio Reporting Summary linked to this article.

## Data availability

The authors declare that all data supporting the findings of this study are available within the article and its Supplementary information files and Source data files or from the corresponding author upon request. The wild-type scRNA-seq data have been previously published[8] and are deposited in the NCBI Gene Expression Omnibus database (GEO) under accession code (GSE158192 [https://www.ncbi.nlm.nih.gov/geo/query/acc.cgi?acc=GSE158192]). The wild-type scATAC-seq data used in the time course analysis are available at GEO (GSE264098 [https://www.ncbi.nlm.nih.gov/geo/query/acc.cgi?acc=GSE264098]). The P3 scATAC-seq data set was previously published[14] and available at GEO (GSM4504962 [https://www.ncbi.nlm.nih.gov/geo/query/acc.cgi?acc=GSM4504962]). The ChIP-seq data is available at the GEO (GSE247271

[https://www.ncbi.nlm.nih.gov/geo/query/acc.cgi?acc=GSE247271]), 10-wk NKX2-1 AT1 ChIP-seq is from our previous publication[8] under GEO (GSE158205 [https://www.ncbi.nlm.nih.gov/geo/query/acc.cgi?acc=GSE158205]). The scMultiome data is available at the GEO (GSE247130 [https://www.ncbi.nlm.nih.gov/geo/query/acc.cgi?acc=GSE247130]). Source data are provided in this paper. Source data are provided with this paper.

## Code availability

A custom script for the analysis and generating the figures is available as part of the Source Data file.

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

## Acknowledgements

We thank Dr. Danielle Little for assistance with ChIP-seq and data analysis and Anne Lynch for assistance with Sendai virus infection. We thank Dr. Harold Chapman for providing the *Sftpc^CreER* mice. The University of Texas MD Anderson Cancer Center DNA Analysis Facility, Flow Cytometry and Cellular Imaging Core Facility, and High-Resolution Electron Microscopy Facility are supported by the Cancer Center Support Grant (P30CA016672). This work was supported by the University of Texas MD Anderson Cancer Center Retention Fund and National Institutes of Health R01HL130129 and R01HL153511 (J.C.).

## Author contributions

D.H. and J.C. designed the research; D.H. performed research and analyzed data; D.H. and J.C. wrote the paper; all authors read and approved the paper.

## Competing interests

The authors declare no competing interests.
