## [Peer Review File · Nature Communications]

CEBPA restricts alveolar type 2 cell plasticity during development and injury-repairREVIEWER COMMENTS

Reviewer #1 (Remarks to the Author):

In this paper, the authors have used single cell sequencing to define differences in AT2 cell populations through aging. Using single cell ATAC sequencing, they found CEBPA as a potential targetable TF in AT1 phenotype. The authors then made a AT2 inducible CEBPA knockout. AT2 CEBPA knockout cells appear to become AT1 cells. They then found that NKX2-1 is co-bound with CEBPA to maintain the AT2 program. Further, CEBPA deletion in mature AT2 cells was unable to reactivate the SOX9 progenitor program. The authors then use Sendai virus infection, which has known effects on AT2 cells, and show that in repair, mature AT2 cells once again gain SOX9 progenitor program as seen in neonatal AT2 cells. Overall the authors have performed very directed and elegant experimentation and have adequately convinced their hypothesis.

Typos

Introduction: stem cells, which secret(e) pulmonary

Results: dependent changes, we FACS purified E-Cadherin+

Higher resolution for figures 1A,B

Figure 1A may be better displayed as a separated panel as time points may sit on top of each other covering the data. As well, it would be good to include a figure displaying the numbers of each cell at each time point.

In both the scRNA and scATAC seq experiments, please state the number of mice used per time point and if there were replicates.

The statement “Eighty-two genes eventually specific to AT2 cells were designated as early versus late AT2 genes based on expression, however low, in >25% of E16.5 SOX9 progenitors” is confusing. Which time points were compared? What was the actual cutoff?

Figure 1C needs a color bar.

For Figure D, what percent of the difference is defined by each axis. Also, is this PC1 and 2?

For Figure D, homer motif analysis, please use corrected p-values and include % background.

Figure S3. Please state the number of mice per experiment and replicates.

For the finding that CEBPA knockout AT2 cells become AT1 cells, the authors should perform more extensive analysis, such as pseudotime analysis, trajectory analysis.... This will allow more confidence with this statement.

The statement “CEBPA-dependent plasticity sites are not bound by CEBPA” is confusing.

Reviewer #2 (Remarks to the Author):

Hassan et al have submitted a detailed analysis of a known AT2 transcription factor, CEBPA, and present insights into its regulation of AT2 cell fate in both the neonatal period and during pulmonary infection. As is currently written and organized, however, some of the conclusions are overstated and some of the

prior works that support this finding are not clearly identified.

In Figure 1, very similar experiments have been published previously (Zepp, et al PMID: 33707239, and Negretti et al PMID: PMID: 34927678) that the authors neglect to comment on how their work adds and / or expands on these findings. Similarly, in figure 2 the authors gloss over the prior work by Martis et al (PMID: 16467360) where that group had eliminated CEBPa in SFTPC+ respiratory epithelium (the specificity of this mouse can be debated, it is limited to the respiratory epithelium and CEBPa expression is limited to AT2 cells) and they also highlight the putative integrations between CEBPa and NKX2-1, which is a cornerstone of this manuscript's findings. The differences between the findings with embryonic development vs post-natal development should be considered, since many of the findings (such as loss of a mature AT2 program as highlighted by the loss of lamellar bodies) is quite similar to the findings in the Martis study. I am not confident that the conclusion that CEBPA promotes AT2 cell fate is really new – but rather this is a more technologically sophisticated approach.

The findings here lead the authors to conclude that CEBPa when lost leads to an emergence of the neonatal SOX9 progenitor fate. As the authors only perform this at P9 and 5w animals, we do not know what the actual switch for “loss of plasticity” is. While the authors don't state clearly that they do know this switch is progenitor potential, they do suggest in the discussion that they have separated the AT2 program specification from ability to de-differentiate – I am not sure that I agree with this conclusion and more data would be needed to truly demonstrate that. The authors would need to identify what is happening that prevents AT2 cells from turning on the embryonic program with age – they could potentially use the time series seq and ATAC data combined with additional KO time points to try to identify when this loss of plasticity occurs and why. What happens at p15? Week 2? 4? And if there is a switch or a gradual loss of this potential? What contributes or accounts for this loss?

The authors use speculative data to claim that CEBPa recruits NKX2-1 based on a reduction in NKX2-1 at concurrent CEBPA CHIP-seq sites. However, the ATAC seq suggests that the region is globally less open which could account for the reduced binding seen in the NKx2-1 CHIP seq. I think this conclusion is overstated.

Figure 5, again, could be more simply interpreted as CEBPa is needed for maintenance AT2 maturation programs which are actively maintained at the transcriptional level – as the crux of the findings are that lamellar bodies are perhaps lost and LAMP3 expression is reduced on IF.

The viral infection study is also problematic in its overstating of conclusions. The use of the progenitor “score” here is the crux of the argument. In the neonatal KO experiment SOX9 is robustly turned on, the cells proliferate and the conclusions are a bit more clear. Here, the SOX9 expression is very limited (both based on 8% pos rate and based on the scRNA shown in fig 6). The Progenitor Score seems to be nearly just as elevated in the control cells as the mutant cells based on the UMAP, and since the Sox9 expression is so limited it is difficult to assess what this means. The fact that a less mature AT2 cell is more likely to become an aberrant DAPT/PATS/ etc like cell is perhaps not surprising. If the authors claim this is a reactivation of the SOX9 progenitor state, they have to do more to prove that functionally – such as trace these cells and show they can give rise to AT2 and AT1 cells, or sort them out (I am not sure how they would sort just those out) and show their potential in an organoid culture ex-vivo for example.

In summary, I think that while the data are interesting, and the controls and analytic approach are valid, that much of the interpretations and conclusions are overstated, which limits my enthusiasm of the manuscript.

Reviewer #3 (Remarks to the Author):

In this paper, Hassan and Chen examined genetic and structural variations within AT2 lung cells across their developmental stages, maturation, and response to viral damage. They observed that CEBPA, an AT2-specific transcription factor, cooperates with NKX2-1 to establish and maintain the identity of AT2 cells. Neonatal AT2 cells exhibited a progenitor program in the absence of CEBPA, extending to mature cells post-viral injury. Viral injury triggered a transformation of AT2 cells into transitional cells expressing KRT8/CLDN4, particularly intensified in the absence of CEBPA. The study highlights CEBPA's pivotal role in AT2 cell development and regeneration at the transcriptional and chromatin structure levels. While the experimental data and analysis are comprehensive, improvements are suggested, particularly in addressing specific comments provided.

Major comments

1. In the second paragraph of the result section, the authors have stated that each category of ATAC peaks (lost and gained groups, each with early and late subgroups), appeared near specific genes in each case. For instance, "As shown in Fig 1D, the early lost peaks....., were near the progenitor and AT1 genes (Adamts18 and Pdpn, respectively)". However, this point is not shown in Fig 1D. Authors can add some data such as snapshots of genomic loci of representative peaks to aid the reader's visual understanding.
2. In Fig 2A, the authors stated that the protein expression of CEBPA in wild-type mice was absent at E16.5 but was expressed from E18.5 in a subset of epithelial cells. However, some previous studies suggest the expression of CEBPA at E16.5 [Martis et al. Development (2006), Berg et al. Am J Physiol Lung Cell Mol Physiol. (2006), Laresgoiti et al. Development (2016)]. Can authors comment on their viewpoint on this?
3. In Fig 2A legend, authors stated " ...CEBPA expression in cuboidal cells outlined with E-cadherin (ECAD)", however, this is unclear from the image. Authors should provide an enlarged view of the region for clarity. Similarly, for Ki67 signal in Fig 2E.
4. In Fig.6, Authors showed that CEBPA-null AT2 cells activate SOX9 expression more frequently than control and tend to undergo self-renew following viral infection. Are these Ki67+ AT2 cells SOX9 positive? It is also important to analyze proliferating AT2 cells in control lung. The author should determine whether Ki67+ AT2 cells are CEBPA negative in infected lung?
5. In discussion and Fig7, it is not clear which molecular event the authors' use of the terms "sea level" and "submerged" reflects in this paper. A more detailed explanation should be added or this part would be deleted. The summary based on the data of the present paper is sufficient in the upper part of Fig. 6. Also, in the DISCUSSION, the description is divided into A and B. In fact, Fig 7 is not divided into A and B.

It should be appropriately revised.

Minor comments

1. In Fig 1C, add a color scale for the heatmap.
2. In Fig 5B, add a scale bar to TEM image and its corresponding measurement in μm in the figure legend.
3. In the Fig 5F legend, "Scale: XX" and "Table XX" should be corrected.
4. The units "ug" and "um" should be corrected to " μg " and " μm ", respectively.

We thank the reviewers for their encouraging feedback and valuable recommendations to further improve our manuscript and have addressed them accordingly below and highlighted our edits with track-change in the manuscript.

REVIEWER #1

Overall the authors have performed very directed and elegant experimentation and have adequately convinced their hypothesis

We thank the reviewer for evaluating our experiments and hypothesis and the positive remarks.

Typos: Introduction: stem cells, which secret(e) pulmonary
Results: dependent changes, we FASC purified E-Cadherin+

We have corrected the typos; thank you.

Higher resolution for figures 1A,B. Figure 1A may be better displayed as a separated panel as time points may sit on top of each other covering the data. As well, it would be good to include a figure displaying the numbers of each cell at each time point.

We have enlarged the plots in Fig. 1A and 1B. As suggested, we have included in Source Data 1 separate panels for each time point with associated cell numbers.

In both the scRNA and scATAC seq experiments, please state the number of mice used per time point and if there were replicates.

ScRNA-seq data was published by our group (Little et al. PMID 33947861) with at least 2 mice for each time point. Methods read "Published scRNAseq dataset (GSE158192)⁸ compiled from 12-time points E14.5, E16.5, E18.5, P4, P6, P7, P8, P10, P15, P20, 10-week-old, and 15-week-old lung was analyzed using Seurat (v4.1)." Fig. 1A legend now includes: "Each time point consists of at least 2 mice profiled as one sample."

Similarly, E16.5, E18.5, P8 and 9-wk scATAC-seq data was generated in house using at least 2 mice each. P3 scATAC-seq data was obtained from publicly available data GSM4504962. Methods read "ScATAC-seq control datasets were from 5 different time points (E16.5, E18.5, P3, P8, and 9-wk). The P3 dataset was obtained from (GSM4504962)." Fig. 1D legends now includes "Each time point consists of at least 2 mice profiled as one sample."

The statement "Eighty-two genes eventually specific to AT2 cells were designated as early versus late AT2 genes based on expression, however low, in >25% of E16.5 SOX9 progenitors" is confusing. Which time points were compared? What was the actual cutoff?

The analysis was detailed in Methods: "AT2-specific genes used in the heatmap were obtained by differential gene expression comparison between 15-wk AT2 and AT1 cells using Findmarkers resulting in 88 distinct genes. Subsequently, the expression level of these 88

genes was compared across different time points: E16.5, E18.5, and 15-wk. Genes were classified as early genes if they were present in at least 25% of E16.5 progenitor cells. The remaining genes were classified as late genes. Within this group, genes were classified as "mature" if the logFC between the 15-week time point and E18.5 exceeded 2 (i.e. 4-fold increase) (Table S1). Out of the 88 gene list, 6 genes were excluded (*Tpt1*, *Mt1*, *Mt2*, *Selenop*, *Wfdc2* and *Ly6e*) as they met both early and mature criteria resulting in 82 genes that was used to generate heatmap in Fig. 1."

We have clarified the Results as "Eighty-two genes specific to 15-week AT2 cells, in comparison to 15-week AT1 cells, were designated as early versus late AT2 genes based on their presence versus absence of expression in >25% of E16.5 SOX9 progenitors (Fig. 1C)."

Figure 1C needs a color bar.

Color bar has been added to Figure 1C.

For Figure D, what percent of the difference is defined by each axis. Also, is this PC1 and 2?

Yes, the axes in Figure 1D represent PC1 and PC2. The percentage of variance is 53% and 19%, respectively. We updated Figure 1D accordingly.

For Figure D, homer motif analysis, please use corrected p-values and include % background. Figure S3. Please state the number of mice per experiment and replicates.

Methods now read "De novo motif analysis was obtained using Homer (v. 4.10) 'findMotifsGenome.pl -size 200 -mask' with random backgrounds, which does not provide corrected p-values".

% background information is now included in Fig. 1D, 3F, 5G.

We assume the reviewer is asking about sc-multiome experiments. Fig. 3A legend now includes "Each sample includes 1 male and 1 female mouse profiled as one sample (same for subsequent sc-multiome experiments)."

For immunostaining in general and in Fig. S3, Fig. 2B legend reads "Images are representative of at least three lungs (same for subsequent immunostainings)."

For the finding that CEBPA knockout AT2 cells become AT1 cells, the authors should perform more extensive analysis, such as pseudotime analysis, trajectory analysis.... This will allow more confidence with this statement.

We have added in Results "RNA velocity analysis confirmed this predicted trajectory bridging AT2 and AT1 cells specifically in the mutant (Fig. S3B)." In Fig. S3B legend: "RNA velocity analysis showing a predicted trajectory from AT2 to AT1 cells through a bridging population specifically in the mutant."

The statement “CEBPA-dependent plasticity sites are not bound by CEBPA” is confusing.

This has been changed to “(CEBPA) does not bind to and thus indirectly represses sites that remain plastic in neonatal AT2 cells.”

REVIEWER #2

Hassan et al have submitted a detailed analysis of a known AT2 transcription factor, CEBPA, and present insights into its regulation of AT2 cell fate in both the neonatal period and during pulmonary infection. As is currently written and organized, however, some of the conclusions are overstated and some of the prior works that support this finding are not clearly identified.

In Figure 1, very similar experiments have been published previously (Zepp, et al PMID: 33707239, and Negretti et al PMID: 34927678) that the authors neglect to comment on how their work adds and / or expands on these findings.

We appreciate the opportunity to clarify that our study not only aligns with but also expands upon these findings. Compared to the Zepp dataset on P3 scATAC-seq, we added 4 other time points, allowing us to identify 4 patterns of lost and gained peaks (Fig. 1D). The Negretti dataset has scRNA-seq up to P14. As noted in Methods, the scRNA-seq data used here has been published by us (Little et al. 2021). The temporal RNA and ATAC data is important to support that “Postnatal transcriptomic and epigenomic maturation of AT2 cells is separate from their embryonic specification”.

We updated the concluding sentence of Fig. 1 as “Taken together, our time-course transcriptomic and epigenomic roadmap of AT2 cell development expand on published datasets^{8,14,15}, highlighting the sequential specification and maturation of AT2 cells and implicating CEBPA in their differentiation and plasticity.”

Similarly, in figure 2 the authors gloss over the prior work by Martis et al (PMID: 16467360) where that group had eliminated CEBPa in SFTPC+ respiratory epithelium (the specificity of this mouse can be debated, it is limited to the respiratory epithelium and CEBPa expression is limited to AT2 cells) and they also highlight the putative integrations between CEBPa and NKX2-1, which is a cornerstone of this manuscripts findings. The differences between the findings with embryonic development vs post-natal development should be considered, since many of the findings (such as loss of a mature AT2 program as highlighted by the loss of lamellar bodies) is quite similar to the findings in the Martis study. I am not confident that the conclusion that CEBPA promotes AT2 cell fate is really new – but rather this is a more technologically sophisticated approach.

We thank the reviewer for drawing attention to the seminal work of Martis et al. and appreciate the opportunity to clarify and distinguish the contributions of our study in the context of existing literature.

The Results now include “Although CEBPA had been shown to promote both AT1 and AT2 cell differentiation in embryonic lungs¹⁶, its role in subsequent AT2 cell maturation and plasticity was unclear.” “Differential expression analysis of *Cebpa* mutant versus control AT2 cells confirmed downregulation of AT2 genes (e.g. *Lyz2*, *Lyz1*, *Sftpb*, and *Il33*) and upregulation of progenitor (e.g. *Sox9*, *Clu*, and *Col18a1*) and AT1 genes (e.g. *Akap5*, *Fbln5*, and *Rtkn2*), although some surfactant genes including *Sftpc* were less reduced – unlike their near-absence upon embryonic *Cebpa* deletion¹⁶, possibly due to RNA perdurance or redundant transcriptional activation (Fig. 3D, S3D, S3E). Relatedly, opposite to the phenotypes reported here and given the AT2-restricted expression of CEBPA (Fig. 2A, S1E, S1F), the defective AT1 cell differentiation in the pan-epithelial embryonic *Cebpa* mutant¹⁶ was likely non-cell autonomous or due to potential toxicity associated with the Cre driver¹⁷.”

The Discussion now includes “Future studies are expected to expand the core NKX2-1/YAP/TAZ/TEAD/CEBPA network to include additional TFs, such as KLF5 and AP-1 members, and effector enzymes, such as histone modifiers and chromatin remodelers²⁸, providing a genome-wide mechanistic understanding of regulators of lung development from genetic studies²⁹.”

The findings here lead the authors to conclude that CEBPa when lost leads to an emergence of the neonatal SOX9 progenitor fate. As the authors only perform this at P9 and 5w animals, we do not know what the actually switch for “loss of plasticity” is. While the authors don’t state clearly that they do know this switch is progenitor potential, they do suggest in the discussion that they have separated the AT2 program specification from ability to de-differentiate – I am not sure that I agree with this conclusion and more data would be needed to truly demonstrate that. The authors would need to identify what is happening that prevents AT2 cells from turning on the embryonic program with age – they could potentially use the time series seq and ATAC data combined with additional KO time points to try to identify when this loss of plasticity occurs and why. What happens at p15? Week 2? 4? And if there is a switch or a gradual loss of this potential? What contributes or accounts for this loss?

We have performed additional experiments and analysis as suggested. “To further define the temporal window of AT2 cell plasticity and potential regulators, we deleted *Cebpa* at additional neonatal time points and found robust SOX9 activation upon deletion at P4, but little SOX9 at P10 (Fig. S4F). Accordingly, comparison of P4 and P10 scRNA-seq of AT2 cells revealed downregulated and upregulated genes that might promote and suppress plasticity, respectively (Fig. S4G). Intriguingly and worthy of future investigation, *Dlk1* was among the most downregulated at P10 and had been implicated in antagonizing Notch signaling and AT2 self-renewal during injury-repair²⁰.”

The authors use speculative data to claim that CEBPa recruits NKX2-1 based on a reduction in NKX2-1 at concurrent CEBPA CHIP-seq sites. However, the ATAC seq suggests that the region is globally less open which could account for the reduced binding seen in the NKx2-1 CHIP seq. I think this conclusion is overstated. Figure 5, again, could be more simply interpreted as CEBPa is needed for maintenance AT2 maturation programs which are actively maintained at

the transcriptional level – as the crux of the findings are that lamellar bodies are perhaps lost and LAMP3 expression is reduced on IF.

We appreciate the thoughtful interpretation of the data. Besides the rationale described in the same paragraph, we added “The parallel changes in both chromatin accessibility and NKX2-1 binding during their developmental gain from progenitors to AT2 cells and loss upon *Cebpa* deletion (Fig. 3F, 4A diagram) does not establish the sequence of events nor rule out the possibility that despite the presence of adjacent CEBPA and NKX2-1 motifs, decreased NKX2-1 binding to these sites was secondary to chromatin closure due to some impact of CEBPA deletion elsewhere in the genome, calling for future systematic deletion of CEBPA binding sites and/or interference of CEBPA binding to them. Relatedly, the recruitment model (Fig. 4A diagram) highlights the new chromatin binding specificity of NKX2-1 conferred by CEBPA, but does not require sequential binding of CEBPA and then NKX2-1 to the chromatin.” We have updated Fig. 4A diagram to highlight the chromatin states as below and added in its legend “Loss of NKX2-1 binding due to *Cebpa* deletion is associated with chromatin closure (open vs closed).”

The viral infection study is also problematic in its overstating of conclusions. The use of the progenitor “score” here is the crux of the argument. In the neonatal KO experiment SOX9 is robustly turned on, the cells proliferate and the conclusions are a bit more clear. Here, the SOX9 expression is very limited (both based on 8% pos rate and based on the scRNA shown in fig 6). The Progenitor Score seems to be nearly just as elevated in the control cells as the mutant cells based on the UMAP, and since the Sox9 expression is so limited it is difficult to assess what this means. The fact that a less mature AT2 cell is more likely to become an aberrant DAPT/PATS/ etc like cell is perhaps not surprising. If the authors claim this is a reactivation of the SOX9 progenitor state, they have to do more to prove that functionally – such as trace these cells and show they can give rise to AT2 and AT1 cells, or sort them out (I am not sure how they would sort just those out) and show their potential in an organoid culture ex-vivo for example.

Thank you for the suggestions. We have provided additional data and further discussed our conclusions in the following ways.

Robustness of progenitor gene activation (statistics in Fig. S6C): “SOX9+ AT2 cells often abutted lobe edges or airways and macro-vessels, topologically distal ends of the respiratory tree favoring de novo growth as we described²¹ (Fig. S5A). The regional preference, in conjunction with localized virus delivery, suggested that the percentage of mutant AT2 cells capable of expressing SOX9 could be much higher. SOX9 activation depended on infection because saline treated control and *Cebpa* mutant lungs did not express SOX9 (Fig. S5B).” “Progenitor score and genes including *Sox9*, *Dlk1*, and *Kif4* were higher in the mutant, despite the said spatial restriction (Fig. 6G, S6C).”

SOX9+ mutant cells are preferentially proliferative. “Proliferative AT2 cell cluster was much more prominent in the mutant, corroborating the KI67 immunostaining, and expressed *Sox9*, suggesting a possible coupling between proliferation and SOX9 activation (Fig. 6F, 6G). Supporting this, despite their low percentage, SOX9+ AT2 cells were more likely to be KI67+ than SOX9- AT2 cells (45% vs 3.5%; 696 SOX9+ cells out of 8562 GFP+ cells from 3 mice; Fig. 6D).”

Discussion of our conclusion: “Despite the transcriptional resemblance, *Cebpa* mutant AT2 cells were not bona fide SOX9 progenitors since without CEBPA, they were not expected to meet the functional definition of differentiation to AT2 cells and they existed in an adult, injured niche different from their native embryonic environment.” In Introduction, we also stated “this study defines cell plasticity as gains in molecular features characteristic of other cell types.”

REVIEWER #3

Major comments

1. In the second paragraph of the result section, the authors have stated that each category of ATAC peaks (lost and gained groups, each with early and late subgroups), appeared near specific genes in each case. For instance, “As shown in Fig 1D, the early lost peaks....., were near the progenitor and AT1 genes (*Adamts18* and *Pdpm*, respectively)”. However, this point is not shown in Fig 1D. Authors can add some data such as snapshots of genomic loci of representative peaks to aid the reader’s visual understanding.

Thank you for the suggestion. We have added representative genomic snapshots of the 4 categories of peaks in Source Data 2.

2. In Fig 2A, the authors stated that the protein expression of CEBPA in wild-type mice was absent at E16.5 but was expressed from E18.5 in a subset of epithelial cells. However, some previous studies suggest the expression of CEBPA at E16.5 [Martis et al. *Development* (2006), Berg et al. *Am J Physiol Lung Cell Mol Physiol.* (2006), Laresgoiti et al. *Development* (2016)]. Can authors comment on their viewpoint on this?

We have added closeup images in Fig. 2A and updated the Results “(CEBPA) had weak diffuse expression in occasional cells at E16.5, likely corresponding to spatially asynchronous onset of alveolar differentiation and consistent with prior reports^{12,13}”.

3. In Fig 2A legend, authors stated “...CEBPA expression in cuboidal cells outlined with E-cadherin (ECAD)”, however, this is unclear from the image. Authors should provide an enlarged view of the region for clarity. Similarly, for Ki67 signal in Fig 2E.

As suggested, we have added closeup images in Fig. 2A and 2E. We also added 60x images in Fig. S1E “showing CEBPA in cuboidal cells outlined by ECAD at E18.5 (arrowhead).”

4. In Fig.6, Authors showed that CEBPA-null AT2 cells activate SOX9 expression more frequently than control and tend to undergo self-renew following viral infection. Are these Ki67+ AT2 cells SOX9 positive? It is also important to analyze proliferating AT2 cells in control lung. The author should determine whether Ki67+ AT2 cells are CEBPA negative in infected lung?

As suggested and now added to Fig. 6D, “Ki67+ cells in the control and *Cebpa* mutant are stratified by CEBPA and SOX9 expression, respectively.” The Results now read “Proliferative AT2 cell cluster was much more prominent in the mutant, corroborating the Ki67 immunostaining, and expressed Sox9, suggesting a possible coupling between proliferation and SOX9 activation (Fig. 6F, 6G). Supporting this, despite their low percentage, SOX9+ AT2 cells were more likely to be Ki67+ than SOX9- AT2 cells (45% vs 3.5%; 696 SOX9+ cells out of 8562 GFP+ cells from 3 mice; Fig. 6D). By comparison, Ki67+ AT2 cells in the infected control lung were equally likely to be CEBPA+ or CEBPA- (5.9% vs 6.0%; 233 CEBPA- cells out of 2642 GFP+ cells from 3 mice; Fig. 6D), suggesting that transition of control AT2 cells into other CEBPA- populations as examined further below was uncoupled from proliferation.”

5. In discussion and Fig7, it is not clear which molecular event the authors' use of the terms "sea level" and "submerged" reflects in this paper. A more detailed explanation should be added or this part would be deleted. The summary based on the data of the present paper is sufficient in the upper part of Fig. 6. Also, in the DISCUSSION, the description is divided into A and B. In fact, Fig 7 is not divided into A and B. It should be appropriately revised.

The sea-level and submerged imagery is to reflect the fact the CEBPA does not bind to plasticity sites and thus must indirectly regulate them, as the sea submerges the landscape without changing it. We have clarified this in Discussion: “Rather than direct binding and regulation by CEBPA, the plasticity landscape is submerged by the AT2 program, partially exposed when altered sufficiently, but unmasked upon *Cebpa* deletion (Fig. 7B)” and in Fig. 7B legend: “Although CEBPA does not bind to genomic regions of plasticity, without CEBPA, the AT2 program subcedes to expose the progenitor program.”

We have added the panel labels A and B to Fig. 7; thank you for catching this.

Minor comments

1. In Fig 1C, add a color scale for the heatmap.

We have added the color scale.

2. In Fig 5B, add a scale bar to TEM image and its corresponding measurement in μm in the figure legend.

We have added a scale bar to the TEM figure and corresponding measurement to the legend.

3. In the Fig 5F legend, "Scale: XX" and "Table XX" should be corrected.

We have corrected the missing information; thank you for the careful read.

4. The units "ug" and "um" should be corrected to " μg " and " μm ", respectively

They have been updated. In the past, journals help update them after acceptance.

REVIEWERS' COMMENTS

Reviewer #1 (Remarks to the Author):

The reviewers have thoroughly responded to all comments. They have included more details of how their work advances the field in relation to previous research.

Reviewer #2 (Remarks to the Author):

I appreciate the time and effort the authors have spent on my recommendations and considerations. I think the manuscript is more in context of prior work that prior and the conclusions are more appropriately stated.

The additional time points for the KO experiments do further support their point and I think make it stronger.

The additions to the viral experiment help, and I understand that add'l functional experiments are probably outside of the scope of this manuscript. This evidence helps support the thesis.

Reviewer #3 (Remarks to the Author):

This reviewer feels all comments have been addressed properly. We do not have any further comments and feel the manuscript is ready for acceptance.